# Functional optic tract rewiring via subtype- and target-specific axonal regeneration and presynaptic activity enhancement

Xin Zhang[1,11], Chao Yang [1,2,3,4,11], Chengle Zhang [1,11], Junqiang Wu[1], Xiang Zhang[5,6], Jiayang Gao[7], Xuejie Wang[1,2], Leung Ting Chan[1], Yiren Zhou[1], Yujun Chen [1], Sindy Sing Ting Tam [1,5], Shuhang Chen[2,5], Yuqian Ma [8], Wing-Ho Yung[9], Liting Duan [10], Liwen Jiang [7], Yiwen Wang[5,6] & Kai Liu [1,2,3,4,6] ✉

Mechanisms underlying functional axonal rewiring after adult mammalian central nervous system (CNS) injuries remain unclear partially due to limited models. Here we develop a mouse intracranial pre–olivary pretectal nucleus (OPN) optic tract injury model and demonstrate that Pten/Socs3 knockout and CNTF expression in retinal ganglion cells (RGCs) promotes optic tract regeneration and OPN reinnervation. Revealed by transmission electron microscopy, trans-synaptic labeling, and electrophysiology, functional synapses are formed in OPN mainly by intrinsically photosensitive RGCs, thereby partially restoring the pupillary light reflex (PLR). Moreover, combining with Lipin1 knockdown accelerates the recovery and achieves functional reconnection after chronic injury. PLR can be further boosted by increasing RGC photosensitivity with melanopsin overexpression, and it can also be enhanced by treatment of a voltage-gated calcium channel modulator to augment presynaptic release. These findings highlight the importance of neuronal types and presynaptic activity for functional reconnection after CNS injuries.

Injuries to the adult mammalian central nervous system (CNS), such as spinal cord injury and optic nerve injury, often result in persistent functional deficits. These deficits arise from the inability of injured axons to regrow and regain their lost function[1–5]. To achieve a successful functional recovery, two crucial yet distinct components must be addressed: axon regeneration and connection re-establishment[6–9]. Considerable efforts have been dedicated to understanding the mechanisms of axon regeneration failure in the CNS, which in turn led to the development of various methods aimed at enhancing axon regeneration by manipulating cell signaling pathways or growth

[1]Division of Life Science, State Key Laboratory of Molecular Neuroscience, The Hong Kong University of Science and Technology, Hong Kong, China. [2]Hong Kong Center for Neurodegenerative Diseases, Hong Kong, China. [3]Biomedical Research Institute, Shenzhen Peking University–The Hong Kong University of Science and Technology Medical Center, Shenzhen, China. [4]Guangdong Provincial Key Laboratory of Brain Science, Disease and Drug Development, HKUST Shenzhen Research Institute; Shenzhen-Hong Kong Institute of Brain Science, Shenzhen, Guangdong, China. [5]Department of Electronic and Computer Engineering, The Hong Kong University of Science and Technology, Hong Kong, China. [6]Department of Chemical and Biological Engineering, The Hong Kong University of Science and Technology, Hong Kong, China. [7]School of Life Sciences, Centre for Cell & Developmental Biology and State Key Laboratory of Agrobiotechnology, The Chinese University of Hong Kong, Hong Kong, Sha Tin, China. [8]Hefei National Research Center for Physical Sciences at the Microscale, CAS Key Laboratory of Brain Function and Disease, Biomedical Sciences and Health Laboratory of Anhui Province, School of Life Sciences, Division of Life Sciences and Medicine, University of Science and Technology of China, Hefei, China. [9]Department of Neuroscience, City University of Hong Kong, Hong Kong, China. [10]Department of Biomedical Engineering, The Chinese University of Hong Kong, Hong Kong, Sha Tin, China. [11]These authors contributed equally: Xin Zhang, Chao Yang, Chengle Zhang. ✉e-mail: kailiu@ust.hk

machineries[3,4,10–12]. Contrastingly, the process of functional reconnection after CNS injuries remains less understood. How do regenerated axons innervate their original targets? Does it require specific neuronal types? What is the role of neuronal activity in the recovery process? Would these reformed synapses function in the same way as normal synapses do under physiological conditions? Addressing these questions will not only improve our understanding of the underlying principles regulating post-injury axonal rewiring but will also provide guidance for further identifying therapeutic targets and developing intervention strategies.

A critical challenge in investigating functional axon rewiring is dissecting the process with a simple and reproducible animal model. Models utilizing injuries to optic nerves have been widely adopted for studying axon regeneration and reconnection[13–17]. However, assessing the functional implications of retinal ganglion cells (RGCs) axonal regeneration after optic nerve injury is difficult due to several factors. These include the limited number of axons innervating the brain, the presence of the optic chiasm barrier, axon misguidance, the long distances axons must travel, and RGC death occurring months after axotomy[9,18–24]. Several other models, such as the pre-chiasm injury model[9], the pre-superior colliculus (SC) injury model[6], and the pre-pretectum injury model[25], have been reported. However, the pre-chiasm model lacks circadian function, while the pre-SC model necessitates the use of 4-aminopyridine for conduction and the function cannot be sustained. Additionally, the pre-pretectum model is a partial injury to the optic tract, making it hard to distinguish effects between regenerated axons and spared axons. Furthermore, the pre-chiasm model and pre-SC model require the removal of a portion of the cortex to ensure lesion completeness, rendering them operationally complex and challenging for others to adopt. To investigate how regenerated axons reinnervate target neurons and regain function, an ideal adult injury model should possess certain characteristics. It should be operationally simple, relying on well-established neuronal circuitry. Furthermore, it should allow for easy confirmation of complete injury and quantitative functional assays, enable the identification of neuron types involved, and result in a straightforward topographic projection to the target nucleus.

In this work, we establish an intracranial pre-olivary pretectal nucleus (OPN) optic tract injury (OTI) model in mice to interrogate functional axonal rewiring. This pre-OPN OTI model involves crushing the bilateral optic tract between the lateral geniculate nucleus (LGN) and OPN/SC, obviating the need to remove large cortex tissue. By activating the intrinsic growth mechanism of RGCs with Pten/Socs3 knockout and CNTF expression[23], we validate that the regenerating axons cross the lesion site and regrow into the OPN. These axons establish new functional synaptic connections with OPN neurons, as demonstrated by electron microscopy, trans-synaptic labeling, and electrophysiology, which partially restored pupillary light reflex (PLR) function. Most of the new functional synaptic connections originate from intrinsically photosensitive RGCs (ipRGCs). Inducing axon regeneration specifically in ipRGCs also leads to partial PLR recovery, indicating that ipRGCs regenerate back to their original targets through short distances. Furthermore, combining Pten/Socs3 knockout and CNTF expression with Lipin1 knockdown accelerates PLR recovery from 6 months to 3 months. To mimic a chronic injury scenario, we initiated this combined treatment 6 weeks after the injury and observed regenerated axons regrowing into the OPN area, leading to partial PLR recovery. We also discover that increasing the photosensitivity of RGCs by overexpressing photopigment melanopsin[26,27] improves PLR function. Notably, single-cell RNA-Seq results reveal that presynaptic release machinery is down-regulated in regenerated RGCs, and increasing synaptic transmission by targeting presynaptic voltage-gated calcium channels with R-roscovitine[28–30] further enhances PLR. Our results show that the pre-OPN OTI model can be a powerful tool to investigate mechanisms underlying post-injury axonal rewiring. Using this model, we provide insights into the critical roles of neuronal types and synaptic functionality for functional reconnection after CNS injuries, generating guidance for enhancing functional recovery mediated by different types of axonal regrowth after varied CNS injuries.

## Results

### Development of an intracranial pre-OPN optic tract injury model

To facilitate the mechanistic investigation and quantitative evaluation of functional axonal rewiring, we developed an intracranial OTI mouse model called "pre-OPN model", where RGC axons are crushed before they reach the OPN and other brain nuclei beyond the LGN. In intact mice, the axons of RGCs extend from the retina into the brain, sequentially forming the optic nerve, the optic chiasm, then through the optic tract to reach different brain nuclei, including LGN, OPN, SC, and others (Fig. 1A, B, and Supplementary Fig. 1A). Our pre-OPN model crushes the optic tract right after LGN, injuring the RGC axons between LGN and OPN/SC (Fig. 1A, B, and supplementary movie 1). At this crushing point, the optic tract is much narrower than in other regions such as SC, not only allowing for easier operation and less brain damage but also avoiding the removal of any cortex tissues. Therefore, compared to the technically challenging pre-chiasm and pre-SC lesion models that require suctioning brain tissue[6,9], the pre-OPN lesion is surgically more convenient and can be readily adopted by other laboratories. In addition, a successful injury can be assessed by the complete cessation of OPN-mediated PLR, a well-established circuit that relies on a specific subtype of RGC, the ipRGCs[31–33]. Thus, the simple and precise measurement of PLR permits the quantitative evaluation of functional axon rewiring as well as the dissection of the role of ipRGCs (Fig. 1C). To completely eliminate the optic tract-mediated PLR function, we performed a bilateral lesion since OPN is innervated by both eyes. The completeness of the bilateral lesion was confirmed by the total loss of PLR function (Supplementary Fig. 1B, C and supplementary movie 2). We also verified the completeness of the lesion by performing a two-color CTB injection. We injected cholera toxin subunit B conjugated with FITC (CTB-FITC) into the vitreous body of the eye before the OTI and then injected CTB-555 intravitreously after the OTI. We analyzed the two-color CTB labeling in the OPN, the nucleus of the optic tract (NOT), the medial division of the posterior pretectal nucleus (mdPPN), and the SC, which are located behind the lesion site. The results demonstrated that there was no CTB-555 signaling in these nuclei that had been pre-labeled with CTB-FITC, as observed in both coronal and sagittal sections (Supplementary Fig. 1D, E). The histological and behavioral assessments showed that the overall success rates of our OTI model are over 89%. We also performed retrograde labeling by injecting CTB-555 into the OPN and SC of intact mice (Supplementary Fig. 2A–C), which are located behind the lesion site, to quantify the minimum proportion of RGCs injured by the OTI. CTB-555 labeled approximately 91% of RGCs, indicating that over 90% of RGCs project their axons behind the lesion site and were injured in the OTI model (Supplementary Fig. 2D, E).

### Regeneration and reinnervation of RGC axons into OPN after intracranial pre-OPN OTI

Next, using our model, we set out to examine whether RGC axons can regenerate beyond the optic tract lesion site to reinnervate the brain nuclei. AAV2 with hSyn promoter was used for targeted transduction of RGCs. To elicit regeneration of RGC axons (denoted as Regen. group), adult *Pten[f/f];Socs3[f/f]* mice received intravitreal injection of AAV2-Cre to knock out Pten/Socs3 as well as AAV2-CNTF to express CNTF, where Pten/Socs3 knockout combined with CNTF expression has been previously shown as a potent treatment for axon regeneration[6,9,23] (Fig. 1D). As an injury control (Injury Ctrl.), another group of mice was subject to the intravitreal injection of AAV2-LacZ. Four weeks after AAV injection, bilateral pre-OPN OTI was performed (Fig. 1D). At 6 months post-injury,

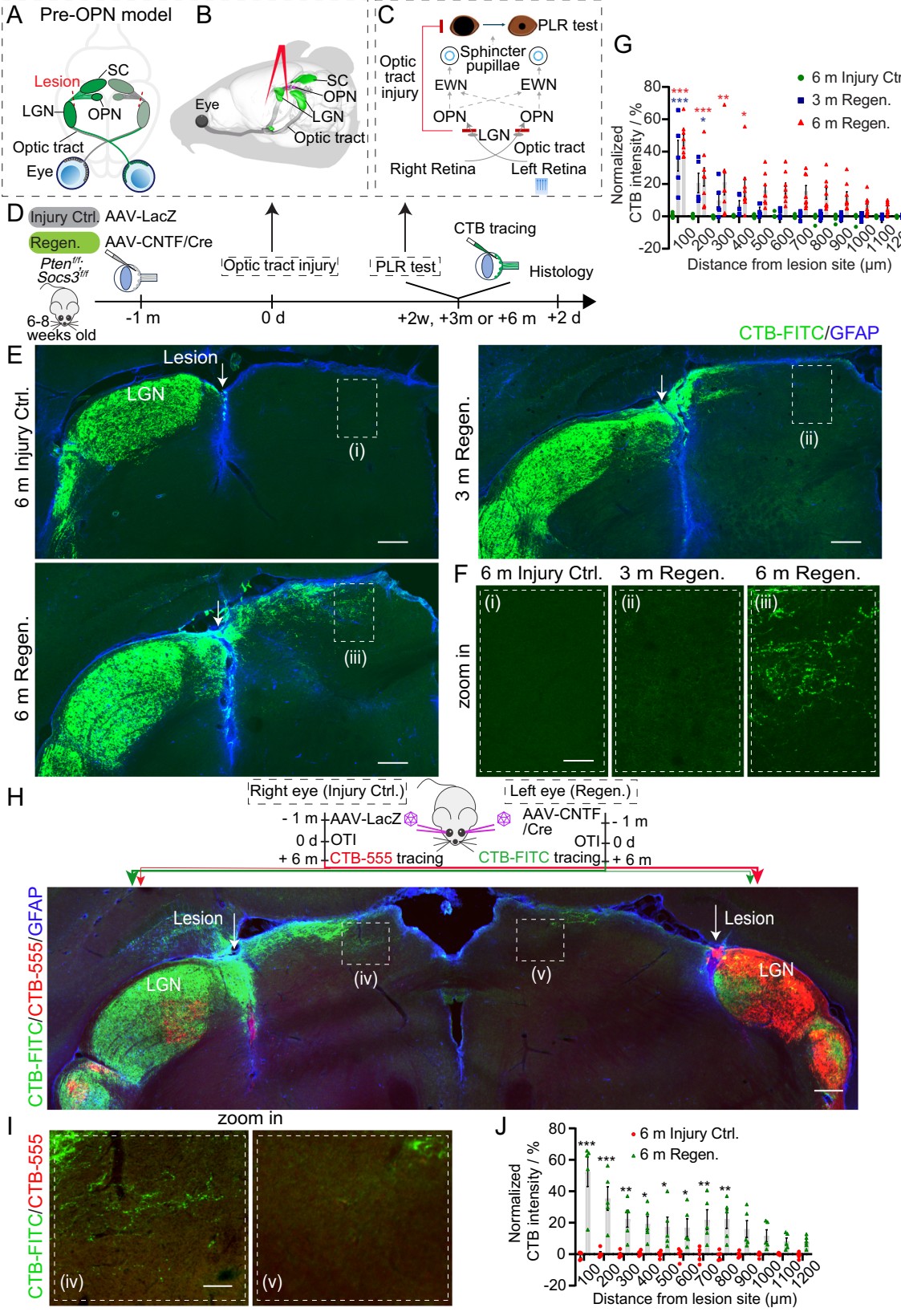

in Injury Ctrl. mice, few axons regenerated beyond the lesion site, as evidenced by CTB-FITC labeling of RGC axons (Fig. 1E−G and Supplementary Fig. 3A). Conversely, at 3 months post-injury in Regen. mice, many RGC axons had extended beyond the lesion site but did not reach the OPN region (Fig. 1E−G). However, in Regen. mice 6 months post-injury, many axons grew much longer and successfully reached the OPN (Fig. 1E−G, Supplementary Fig. 3B, C, and Supplementary Fig. 11A), reinnervating both the shell and core regions of OPN, as delineated by parvalbumin staining in accordance with previous reports[34–36] (Supplementary Fig. 4A). The distribution of these regenerated RGC axons

**Fig. 1 | Design of an intracranial pre−OPN OTI model and the regeneration and reinnervation of injured RGC axons into OPN after Pten/Socs3 knockout and CNTF expression. A** Schematic illustration of the pre-OPN bilateral OTI model where RGC axons are crushed at the narrow region of the optic tract right after LGN, between LGN and OPN/SC. **B** Schematic illustration of performing OTI on the optic tract after LGN without removing cortical tissues. **C** Schematic diagram showing the PLR pathway. Bilateral OTI results in a complete cessation of OPN-mediated PLR. **D** Schematic diagram illustrating the experimental timeline for (**E−G**). **E** Fluorescence images showing CTB-labeled RGC axons (green) and GFAP-labeled lesion site (blue) in LGN and OPN from different groups. Scale bar: 200 μm. **F** Zoomed-in images of rectangles-indicated areas in (**E**) showing RGC axons in the OPN regions. Scale bar: 50 μm. **G** Quantification of the normalized CTB intensity at different distances from the lesion site. Blue/Red asterisks indicate the comparisons between 3 m Regen. group and 6 m Injury Ctrl. group, and 6 m Regen. group

vs. the 6 m Injury Ctrl. group respectively. Data presents mean ± SEM. n = 7 (6 m Injury Ctrl.), n = 5 (3 m Regen.), and n = 8 (6 m Regen.). Statistical significance: Red asterisks p < 0.0001 (100 and 200 μm), p = 0.0014, 0.0499 (300, 400 μm). Blue asterisks p < 0.0001 (100 μm), p = 0.0295 (200 μm). **H** Fluorescence images showing the CTB-FITC (green) labeled RGC axons from the left Regen. eye and CTB-555 (red) labeled RGC axons from the right Injury Ctrl. eye from the same mouse. Scale bar: 200 μm. **I** Zoomed-in images of rectangles-indicated areas in (**H**) showing RGC axons in the OPN regions. Scale bar: 50 μm. **J** Quantification of the normalized CTB intensity in the contralateral hemisphere of the injected eye at indicated distances from the lesion site. Data (mean ± SEM): n = 5 mice. Statistical significance: p < 0.0001 (100 and 200 μm), p = 0.0025, 0.0227, 0.0403, 0.0467, 0.0027, 0.0018 (300-800 μm). * p ≤ 0.05, ** p ≤ 0.01, *** p ≤ 0.001. ANOVA followed by Bonferroni test, two-sided (**G, J**). Source data are provided as a Source Data file.

between LGN and OPN was more diffusive, and some of them grew dorsally toward the cortex or ventrally toward the thalamus, representing a typical pattern for regenerated axons contrasting with the focused distribution observed in intact mice (Fig. 1E, Supplementary Fig. 1A, Supplementary Fig. 3B, and Supplementary Fig. 4B). Additionally, we identified side branches from the regenerated axons, indicating the presence of axon collaterals (Supplementary Fig. 4C). We also assessed the regeneration intensity of axons in the OPN by counting the number of CTB-labeled axon terminal boutons in the OPN for the 6 m Regen. group and the sham control (Sham). The Sham mice had approximately 90 boutons per 100 μm² in the OPN, while the 6 m Regen. group had about 10 boutons per 100 μm² (11% of the CTB boutons in Sham mice) (Supplementary Fig. 4D, E).

To further illustrate the utility of the pre-OPN model in scrutinizing the regeneration and reinnervation process of RGC axons, we conducted another experiment where, in the same mouse, one eye was injected with AAV-Cre and AAV-CNTF, serving as the regeneration group, while the other eye received AAV-LacZ injection, acting as the internal injury control (Fig. 1H). Bilateral pre-OPN OTI was performed 1 month after AAV injection. At 6 months post-injury, RGC axons from the Regen. eye were traced by CTB-FITC and those from the Injury Ctrl. eye were labeled by CTB-555, allowing color-based differentiation between the two. Many axons from the Regen. eye progressed beyond the lesion site, reinnervating the OPN regions on both sides (Fig. 1H−J). Opposingly, RGC axons from the Injury Ctrl. eye did not show any regeneration beyond the lesion site, nor did they reach the OPN regions on either side (Fig. 1H−J).

Using the pre-OPN model, we also discovered that intracranial OTI did not cause noticeable RGC death, in stark contrast with the substantial RGC loss caused by the intraorbital optic nerve crush (ONC)[11,37] (Supplementary Fig. 5A, B). The striking difference in RGC survival is only partially attributable to the number of RGCs that are injured, as the OTI model affects over 90% of RGCs, according to the CTB-555 retrograde labeling results (Supplementary Fig. 2D, E). Instead, the distance from the lesion site and the presence of intact branches before the lesion site appear to be more significant factors, as previously reported[18,38–40]. However, their influence on the RGC response requires further investigation at the molecular level to provide a comprehensive understanding of RGC survival and injury. Inspired by the differing results induced by OTI and ONC, we conducted bulk RNA sequencing to compare the transcriptomic responses of RGCs to these two injury models. This revealed slower and weaker transcriptional changes (Supplementary Fig. 5C) as well as fewer activations of genes related to cell death pathways (Supplementary Fig. 5D) after OTI. Given the role of DLK as a critical retrograde injury signal for axotomy-induced RGC death[41,42], we further analyzed the transcription of DLK-dependent pro-apoptotic genes. Activation of these genes was relatively limited following OTI, unlike the enhanced expression seen after ONC (Supplementary Fig. 5E, F). Experimentally validating this intriguing trend, immunostaining of RGCs in retinas revealed much lower levels of p-c-

Jun, a downstream target of DLK, after OTI compared to ONC (Supplementary Fig. 6A, B). The vast majority of activated p-c-Jun (98%) following OTI was found in the RGCs affected by the surgery (Supplementary Fig. 6C, D). These data suggest that minimal RGC death after OTI is associated with slow and weak DLK signaling activation.

Taken together, these results demonstrate the effectiveness of the pre-OPN model in assessing the regeneration and reinnervation processes of RGC axons.

## Regenerated axons reform synapses in OPN

We proceeded to investigate whether the regenerated axons could form synaptic connections with neurons in the OPN. The Regen. and Injury Ctrl. groups were prepared as in Fig. 1D. Six months after injury, regenerated axon terminal boutons were labeled by CTB, with co-staining of presynaptic marker Bassoon and postsynaptic marker Homer1 in OPN (Fig. 2A). The close proximity of CTB, Bassoon and Homer1 can indicate the existence of synapses. Using super-resolution microscopy, many pairs of specks exhibit triple staining at the OPN of Sham or Regen. mice were observed, while none was seen in Injury Ctrl. mice (Fig. 2B, Supplementary Fig. 7A-7B). Although the number of CTB-labeled axon terminal boutons in the OPN for the 6 m Regen. group is about 11% of the CTB boutons observed in Sham mice (Supplementary Fig. 4D, E), the percentage of RGC axon terminal boutons forming synapses is comparable between the two groups (Fig. 2C). This finding indicates that regenerated axons can successfully reestablish synaptic connections in the OPN.

Next, we used transmission electron microscopy (TEM) to further examine synapses in OPN formed by RGC axons, which can be indicated by RGC mitochondria localized near synaptic-vesicles (SV) and postsynaptic density (PSD)-enriched synaptic structures. Mitochondria in RGC axons were labeled with COX4-dAPEX2 via intravitreal injection of AAV-COX4-dAPEX2[43]. The dAPEX2, following peroxidase reaction and osmium tetroxide (OsO₄) treatment, provided high TEM contrast to the labeled mitochondria, enabling clear differentiation of the dark RGC mitochondria from other unlabeled, light-colored mitochondria (Fig. 2D). In both Sham and Regen. mice, dark-colored mitochondria, and SV were observed in the presynaptic region, with discernible synaptic clefts between the pre- and postsynaptic areas and PSD (Fig. 2E). Oppositely, these features were absent in Injury Ctrl. mice. No significant differences were noted in the number of SV or the length of PSD between the Sham and Regen. groups. (Fig. 2F). Our results showed that after pre-OPN OTI, regenerated RGC axons formed synapse structures in the OPN.

## Synapses formed by regenerated RGC axons after pre-OPN OTI are functional

We investigated whether synapses formed by regenerated axons were functional by employing two methods: trans-synaptic tracing and in vivo electrophysiology. Firstly, we used a recombinant Bartha strain of pseudorabies virus (PRV) expressing GFP (PRV-GFP) for multisynaptic

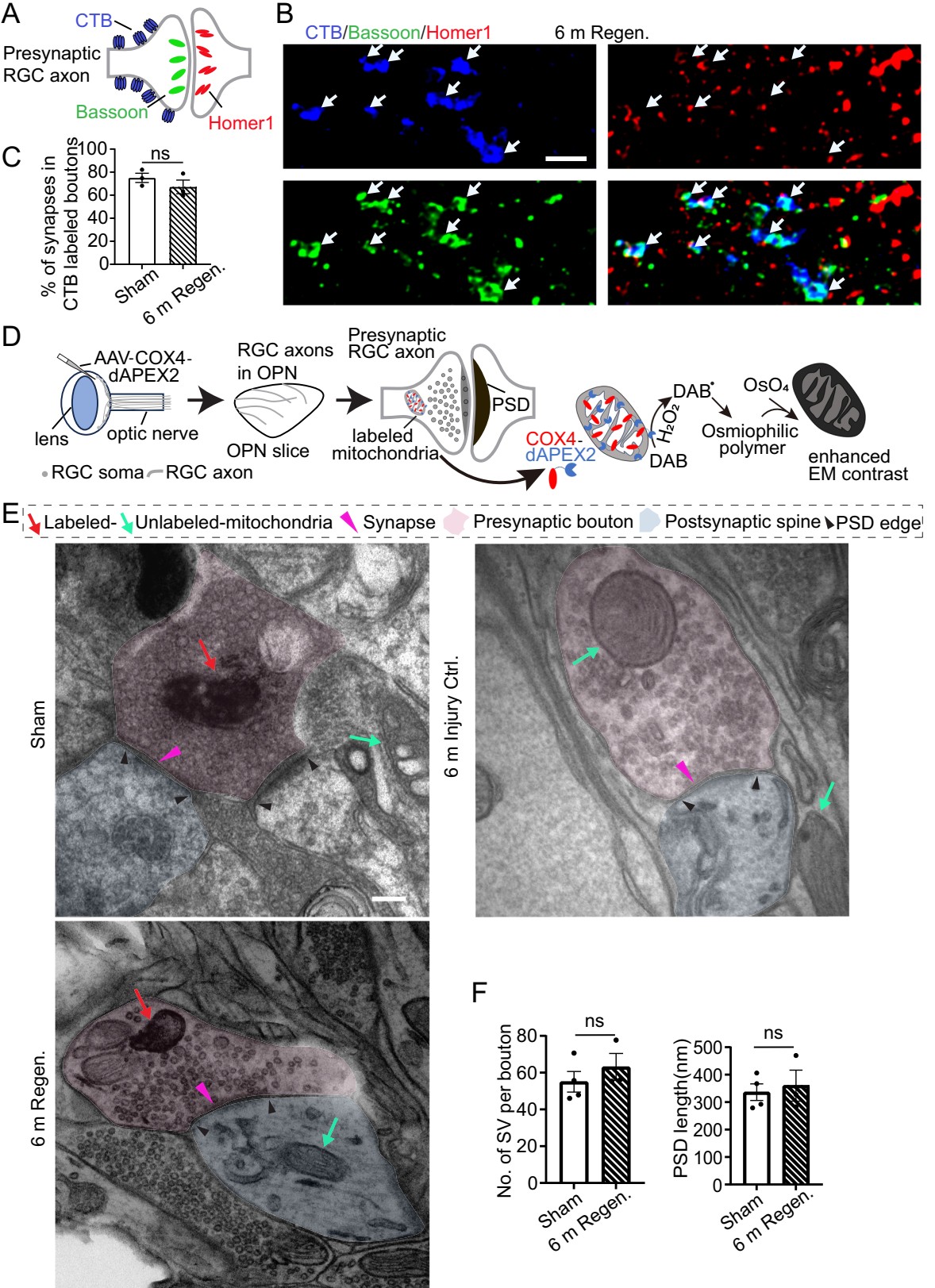

retrograde tracing[44,45], and a glycoprotein-deleted rabies virus expressing dsRed (RV-ENVA-ΔG-dsRed) for monosynaptic retrograde tracing[46]. In the multisynaptic tracing experiment, after the injection of PRV-GFP into the anterior chamber of one eye, it passes through the ciliary body to retrogradely infect ciliary ganglion cells, then reaching the Edinger-Westphal nucleus (EW) and OPN. The EW receives afferents

from the OPN and is involved in the PLR circuit[47,48]. RGCs from the other eye, whose axons establish synapses with OPN neurons that project to the EW, can be labeled with GFP (Fig. 3A). PRV-GFP was injected into Sham mice as well as Injury Ctrl. or Regen. mice 6 months after the pre-OPN OTI. In Sham mice, an average of 418 GFP-labeled RGCs per retina were detected 5 days after PRV injection, predominantly located on the

**Fig. 2 | Synapse formation in OPN by regenerating RGC axons 6 months after pre-OPN OTI. A** Schematic diagram showing the labeling of synapses formed by RGC axons (anterogradely traced with CTB, blue) with the presynaptic marker (Bassoon, green) and postsynaptic marker (Homer1, red). **B** Fluorescence images showing the colocalization of CTB (blue) and Bassoon (green), with Homer1 (red) being adjacent to Bassoon, indicating synapses formed by regenerated RGC axons (marked by white arrows) in the OPN of 6 m Regen. mice. Scale bar: 2 μm. **C** Quantification of the percentage of synapses in CTB-labeled boutons in the Sham and 6 m Regen. groups. Data presents mean ± SEM. n = 3 mice. Statistical significance: p = 0.6362 student's t-test, two-sided. **D** Schematic illustration of identifying dAPEX2-tagged dark mitochondria from presynaptic RGC axons in OPN with TEM. The mitochondrial matrix of the RGC axon terminal was tagged with COX4-dAPEX2 via AAV-COX4-dAPEX2 intravitreal injection, where dAPEX2 labeled mitochondria show high contrast after peroxidase reaction and OsO4 treatment. **E** TEM images of OPN slices from different groups showing dAPEX2-labeled mitochondria in presynaptic RGC axons (indicated by red arrows), unlabeled mitochondria from other regions (indicated by green arrows), synapses (indicated by purple triangles), and the edge of PSD (indicated by black arrows). The pseudo-light pink color represents the presynaptic area, while the light blue color represents the postsynaptic area. Scale bar: 200 nm. **F** Quantification of the number of synaptic vehicles (SV) per bouton (left panel) and PSD lengths (right panel). Data presents mean ± SEM. n = 4 (Sham) and n = 3 (6 m Regen.). Statistical significance: p = 0.7936 (left panel), p = 0.4446 (right panel) Student's t-test, two-sided. ns, not significant, p > 0.05 Source data are provided as a Source Data file.

ventral side (Fig. 3B-3C). On the contrary, no GFP-labeled RGC was seen in the retina of Injury Ctrl. mice, indicating again the complete injury in the pre-OPN OTI model and demonstrating the specificity of the PRV-GFP method in evaluating the levels of synapse formation between RGCs and OPN neurons. In the retina of Regen. mice, the average number of labeled RGCs was 50 (~12% of PRV-labeled RGCs in Sham), and they dispersed throughout the retina with no discernible pattern (Fig. 3B, C). Moreover, we examined the GFP-labeled neurons in the brains of Injury Ctrl. mice 5 days following PRV injection and found that GFP was present in the EW and OPN, which are located downstream of the lesion site in the PLR circuit, but not in the LGN and suprachiasmatic nucleus (SCN) regions, which are upstream of the lesion site (Supplementary Fig. 8A–E). This indicates that the GFP expressed in RGCs in the Regen. group results from trans-synaptic transport between OPN neurons and regenerated RGC axons within the OPN. For the monosynaptic tracing experiment, AAV-Helper (AAV-DIO-EGFP-TVA, AAV-DIO-RVG) and AAV-Cre were injected into the OPN of Sham mice, Injury Ctrl. or Regen. mice 6 months after the pre-OPN OTI. Following this, RV-ENVA-ΔG-dsRed was injected into the same OPN regions for monosynaptic retrograde tracing of RGCs that form synapses with OPN neurons (Supplementary Fig. 9A–F). Consistent with the multisynaptic tracing results, no dsRed-labeled RGCs were found in the retinas of Injury Ctrl. mice (Supplementary Fig. 9G, H), further confirming the completeness of the injury and the specificity of the rabies virus infection. In Sham mice, an average of 1304 dsRed-labeled RGCs per retina were identified, whereas the average number of labeled RGCs in the Regen. mice were 89 (Supplementary Fig. 9H). These results show that regenerated axons not only physically arrived at OPN but also established synaptic connections with OPN neurons.

Secondly, we assessed the functionality of the reformed synaptic connections via in vivo electrophysiology recording (Fig. 3D). If RGCs are functionally connected with OPN neurons, light stimulation received by RGCs can trigger neuronal activities of OPN neurons at the contralateral side (Supplementary Fig. 10A), which can be indicated by the number of spikes per second (firing rate) measured by an implanted microelectrode array in this OPN. To confirm that the implanted electrode specifically measured OPN activities, AAV-hM3D(Gq) was injected into the OPN of Sham mice and indeed systemic administration of CNO elicited a partial PLR response (44%) (Supplementary Fig. 10B) and increased the neuronal firing of OPN neurons (Supplementary Fig. 10C). In Sham mice, light stimulation significantly increased OPN neuronal firing rate (Fig. 3D). In contrast, light stimulation failed to increase neuronal firing rate in the OPN of the 6 m Injury Ctrl. mice. However, light significantly boosted the neuronal responses in 6 m Regen. mice, although not as strongly as in the Sham condition (Fig. 3D and Supplementary Fig. 10D), suggesting the existence of functional synaptic connections between the regenerated RGC axons and OPN.

Therefore, both the trans-synaptic tracing assay and the in vivo electrophysiology experiment jointly demonstrated that the synapses formed by regenerated RGC axons and OPN neurons were indeed functional.

## Regenerated axons partially restore pupillary light reflex after optic tract injury

We next examined whether the reformed synapses at OPN could restore PLR in mice. We assessed PLR using continuous 0.5 mW/cm$^2$ light stimulation at 480 nm for one minute (Fig. 3E). In Sham mice, light stimulation induced pupil constriction to ~90% (Supplementary Fig. 1B, C). In Injury Ctrl. mice up to 6 months post-injury or in Regen. mice up to 3 months post-injury, no PLR response was observed; pupils remained unresponsive to the light stimulation. On the contrary, 6 months post-injury in Regen. mice, 0.5 mW/cm$^2$ light stimulation resulted in an average of 27% pupil constriction (Fig. 3E, F, Supplementary Fig. 11B, and supplementary movie 3), demonstrating that PLR was partially restored by post-injury regeneration and functional reconnection of RGC axons. Further analysis of the kinetics of light-induced pupil constriction at 6 months post-injury revealed that it took 60 s to reach maximum constriction in Regen. mice, compared to only 20 s in Sham mice (Fig. 3G). A similar delay was observed when the demyelination agent Lysophosphatidylcholine (LPC)[49] was injected into the optic tract of Sham mice (Supplementary Fig. 11C, D), suggesting that myelination plays a critical role in the speed of pupil constriction response dynamics, although the maximum constriction was not affected (Supplementary Fig. 11C). In addition, we found that higher light intensity was required to trigger noticeable PLR in Regen. mice than in Sham mice (Fig. 3H). Specifically, light stimulation as strong as 8 mW/cm$^2$ caused 96% pupil constriction in Sham mice but only 29% in Regen. mice (Fig. 3H). Intriguingly, the majority of ipRGCs, labeled by GFP in the Opn4-GFP mouse (4%) and responsible for mediating PLR function, were unmyelinated (Supplementary Fig. 11E–G). Nevertheless, our results show that the reformed synapses at OPN could partially restore PLR.

## ipRGCs regenerate their axons to mediate the PLR response after injury

Then we asked whether specific types of RGCs contributed to the functional rewiring after injury. In contrast to the monosynaptic rabies virus tracing, which labels all RGCs that reestablish synapses with OPN neurons regardless of their relation to the PLR response, multisynaptic retrograde tracing using PRV-GFP can only label RGCs whose axons reestablish synapses with OPN neurons projecting to the EW. This is because PRV-GFP was delivered to the anterior chamber of the eye to infect the iris, which is downstream of OPN in the PLR circuit. Thus, we further analyzed the RGC subtypes labeled by PRV-GFP in the retina. Given that pupil constriction in uninjured conditions is mediated by OPN-innervating ipRGCs[31,32], we used a melanopsin antibody to mark M1-M3 ipRGCs and an additional SMI32 antibody to mark alpha RGCs. In Sham mice, we detected melanopsin + ipRGCs, SMI32+ alpha RGCs, and unidentified other types of RGCs among PRV-labeled RGCs. However, in Regen. mice, only melanopsin + ipRGCs and SMI32+ alpha RGCs were labeled by PRV-GFP (Supplementary Fig. 8F). In terms of cell type percentages, about 60% of PRV-labeled RGCs were melanopsin+ ipRGCs and 15% were SMI32+ alpha RGCs, while the remaining 25% were other types of RGCs in

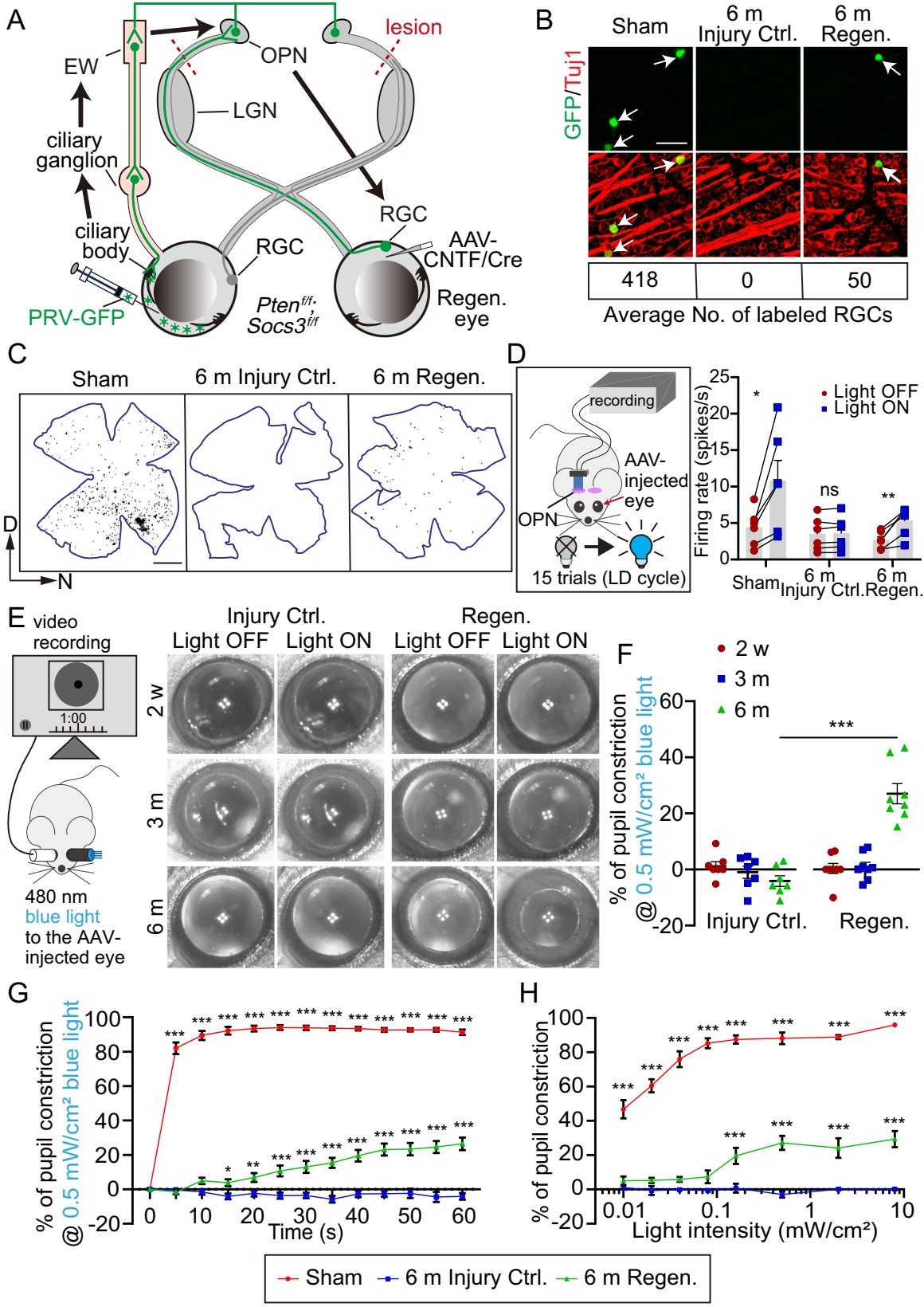

Sham mice (Fig. 4A). In contrast, in Regen. mice, the percentage of melanopsin+ ipRGCs in all PRV-labeled RGCs increased to 88%, with the rest 12% all being SMI32+ alpha RGCs (Fig. 4A, B). These findings suggest that the ipRGC subtype constitutes the majority of RGCs that form synapses with the OPN neurons related to the PLR response in the Regen. mice.

We then investigated whether the axon regeneration of ipRGCs was sufficient to restore PLR. To this end, we performed the optic tract injury in *Opn4*^Cre^;*Pten*^f/f^;*Socs3*^f/f^ mice where Pten/Socs3 knockout was restricted only in ipRGCs (Fig. 4C). Combining Pten/Socs3 knockout and CNTF expression in ipRGCs, none of the animals showed PLR response at 6 months after injury (Fig. 4D), with some axons crossing

**Fig. 3 | Functional synapse formation and partial PLR recovery by Pten/Socs3 knockout and CNTF expression 6 months after pre-OPN OTI. A** Schematic of PRV-GFP retrograde tracing from the contralateral eye's iris to RGCs in the Regen. eye. Arrows indicate the PRV-GFP tracing trajectory with PRV-GFP and the infected neurons labeled in green throughout the process. **B** Fluorescence images showing PRV-GFP-labeled cells (green) co-localized with the RGC marker Tuj1 (red) in different groups 5 days after PRV injection. The lower panel shows the PRV-labeled RGCs from Sham (11 mice), 6 m Injury Ctrl., and 6 m Regen. group (3 mice each). Scale bar: 50 μm. **C** Flat mount retina images showing PRV-traced RGCs (black dots) from different groups. Scale bar: 1000 μm. **D** Schematic of in-vivo electrophysiology experiment, and the quantification of firing rate (spikes/second) before and after blue light (0.5 mW/cm²) stimulation in various groups. Data presents mean ± SEM (n = 6 per group). Statistical significance: p = 0.019, 0.3074, 0.0052 (sham, 6 m Injury Ctrl., 6 m Regen.). Multiple paired t-tests, two-sided. **E** Schematic of PLR testing and representative pupil images before and after 1-minute blue light (0.5 mW/cm²) stimulation. **F** Quantification of percentages of pupil constrictions in different groups at different time points. Data presents mean ± SEM. n = 7 for Injury Ctrl., n = 8 for Regen.. Significance: p ≤ 0.0001, ANOVA followed with Bonferroni test, two-sided. **G** Quantification of the pupil constriction dynamics within 1 min of blue light (0.5 mW/cm²) stimulation in different groups. Data presents mean ± SEM. n = 7 (6 m Injury Ctrl. and Sham), n = 8 (6 m Regen.). Significance: Sham vs. 6 m Injury Ctrl: p ≤ 0.0001 (5-60 s). 6 m Regen. vs. 6 m Injury Ctrl: p = 0.0246, 0.0082 (15 s, 20 s), p < 0.0001 (25–60 s). ANOVA with Dunnett's test, two-sided. **H** Quantification of pupil constriction percentages in response to varying light intensities across groups. Data presents mean ± SEM. n = 6 (Sham and 6 m Injury Ctrl.), n = 7 (6 m Regen.). p ≤ 0.0001 (all marked asterisks), ANOVA with Dunnett's test, two-sided. ns, not significant, p > 0.05, * p ≤ 0.05, ** p ≤ 0.01, *** p ≤ 0.001 Source data are provided as a Source Data file.

the lesion site but not reaching the OPN (Fig. 4E, F). However, 12 months after injury, PLR responses in the regeneration group mice were partially restored, exhibiting 16% pupil constriction upon blue light stimulation (Fig. 4D). Indeed, some axons extended beyond the lesion sites and arrived at the OPN area (Fig. 4E, F). Oppositely, the Injury Ctrl. mice did not show functional recovery or axon regeneration (Fig. 4D–F). Our results demonstrated that ipRGC is crucial for functional reconnection after injury, such that ipRGC regeneration alone can mediate partial PLR recovery.

The majority of ipRGC axons, which are responsible for mediating PLR function, lack myelination. However, we discovered that LPC-induced demyelination did not affect maximum constriction, but the pupillary response was delayed (Supplementary Fig. 11C, D), indicating that myelination is required for a quick PLR response. We observed partial PLR recovery in both AAV and Opn4-Cre-induced regeneration mice with slower pupil constriction dynamics than Sham mice. It is most likely because the majority of regenerated axons are unmyelinated, which could explain the slower pupil constriction. Regenerated alpha RGCs may also contribute to PLR, but their function is affected by demyelination. Promoting remyelination of the regenerated alpha RGC axons could potentially accelerate pupil constriction dynamics in the regeneration mice, which requires further investigation.

### Lipin1 knockdown expedites functional recovery by increasing optic tract regeneration

Although we demonstrated that the combination of Pten/Socs3 knockout and CNTF expression effectively promoted RGC axon regeneration and partial functional recovery following pre-OPN OTI, the process took 6 months, posing significant challenges for mechanistic studies (Fig. 1E–G and Fig. 3E, F). Therefore, we explored whether additional treatments could expedite RGC axon regeneration and functional recovery, including overexpression of cMyc or MEK, or knockdown of GSK3b or Lipin1[50–54]. Among these approaches, the combination of Lipin1 knockdown with Pten/Socs3 knockout and CNTF expression (abbreviated as PSCL Regen.) significantly accelerated axon regeneration. Lipin1 is a phosphatidic acid phosphatase (PAP) enzyme that is essential for the conversion of phosphatidic acid to diacylglycerol in the glycerol phosphate pathway[55]. Previous research has demonstrated that the depletion of lipin1 redirects lipid metabolism from triacylglyceride to phospholipid synthesis and enhances the mTOR and STAT3 signaling pathways to facilitate axon regeneration[54,56]. Additionally, axon regeneration was synergistically improved by combining Lipin1 knockdown with PTEN deletion or CNTF overexpression following optic nerve injury[54]. In the pre-OPN OTI model, regenerated axons notably traversed the lesion site and reinnervated the OPN within just 3 months post-injury (Fig. 5A–C). Furthermore, only 3 months after the injury, mice treated with PSCL Regen. displayed a detectable PLR, with an average pupil constriction of 16.5% (Fig. 5D). This finding underscores that strategic combinations

like PSCL can not only hasten axon regeneration but also significantly speed up the recovery of PLR.

### Axon regeneration and functional recovery after chronic optic tract injury

Chronic CNS lesions pose significant challenges for both axonal regeneration and functional recovery. Although axonal regeneration in such contexts has been previously documented[57–59], achieving functional recovery after complete injury remains challenging. In this study, we investigated whether PSCL Regen. treatment could regenerate RGC axons at a chronic stage post-pre-OPN OTI and subsequently facilitate the recovery of PLR. We initiated PSCL Regen. treatment 6 weeks after pre-OPN OTI (Chronic PSCL Regen.) (Fig. 5E). PLR responses were monitored at multiple time points post-injury. Initially, none of the treated mice demonstrated a PLR response to light stimulation (Fig. 5F). However, beginning 5 months post-injury, PLR was detected in one mouse within PSCL Regen. group. By 7 months post-injury, PSCL Regen. mice exhibited an average pupil constriction of 19% in response to light, whereas the Injury Ctrl. mice's pupils remained unresponsive (Fig. 5F and Supplementary Fig. 12A). Corroborating these functional observations, we also found that RGC axons in the PSCL Regen. group had regenerated, crossed the lesion site, and reinnervated the OPN by 7 months post-injury. This regenerative activity was absent in the Injury Ctrl. group (Fig. 5G, Supplementary Fig. 12B, C). Our findings indicate that PSCL Regen. treatment not only promotes axonal regeneration but also contributes to the partial recovery of the PLR, even when administered during the chronic phase of an injury.

### Increasing RGC photosensitivity by melanopsin overexpression enhances PLR recovery

We investigated whether boosting RGC activity could further improve PLR recovery following axon regeneration post-OTI, building on previous discoveries. We and others have demonstrated that enhancing neuronal activity through chemogenetic or optogenetic techniques can promote RGC axon regeneration after optic nerve injury[26,60]. Similarly, rehabilitation training and electrical stimulation have been effective in aiding functional recovery after spinal cord injuries[61–64]. In this work, PSCL Regen. mice underwent 4 weeks of neuronal activity training, by either light/dark alternating stimulation or DREADD-Gq activation, starting from 3 months post-OTI (Supplementary Fig. 13A). For the light/dark training, we tested various durations of light-dark cycles and found that a 1 h light followed by 1 h dark cycle was most effective in increasing neuronal activity, as evidenced by elevated cFos expression in the retina and OPN (Supplementary Fig. 13B). Therefore, 1 h alternating light/dark stimulation was adopted for the 4-week training. Blue light was delivered into the cage at 0.6 mW/cm² (approximately 618 lux), significantly greater than the typical light power (~0.03 mW/cm²). For the DREADD-Gq-based training, mice received AAV-hM3D(Gq) intravitreal injection 2 months after injury,

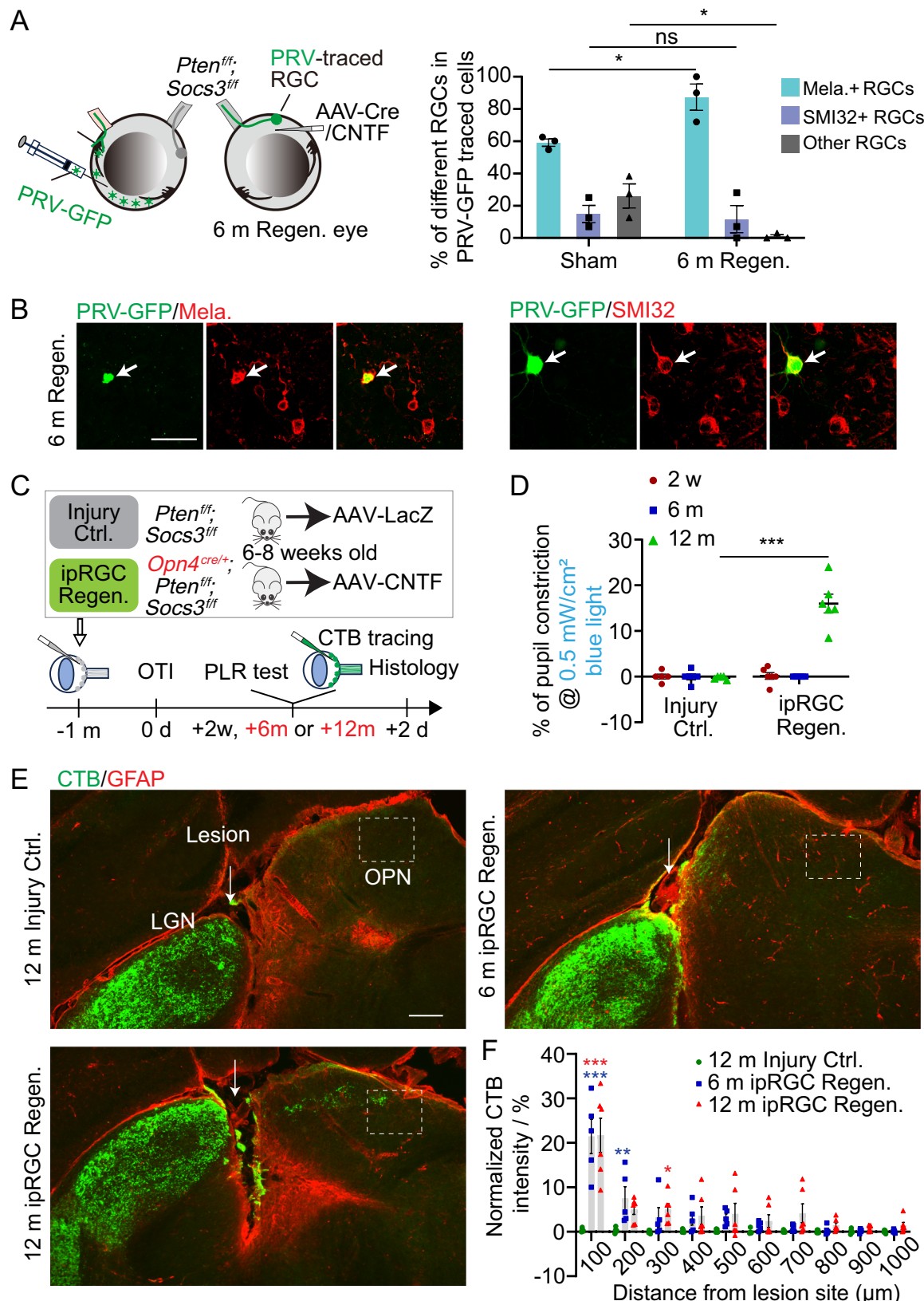

followed by bi-daily intraperitoneal injections of 5 mg/kg CNO starting at 3 months post-injury for one month. PLR was assessed after the 1-month training (Supplementary Fig. 13A). Although we have shown that chemogenetic activation of RGC by DREADD-Gq was very robust by being able to induce a bigger pupil constriction than light stimulation in the same 3 m PSCL Regen. mice (Supplementary Fig. 13D, E),

neither the 1-month light/dark training nor DREADD-Gq training further improved the recovery of PLR (Supplementary Fig. 13C).

However, the strong PLR responses from DREADD-Gq activation suggested that enhanced input to RGCs could further boost PLR. Motivated by this, we explored melanopsin overexpression as another strategy to improve PLR recovery when regeneration is partial.

**Fig. 4 | ipRGCs are the main subtype reforming functional synapses and ipRGCs axonal regeneration partially restores PLR. A** Quantification of the percentages of Melanopsin+ ipRGCs, SMI32+ αRGCs, and other RGC types in PRV-GFP traced cells from the respective groups. Data presents mean ± SEM. n = 3 mice. Statistical significance: p = 0.0216 (Mela+ RGCs), p > 0.999 (SMI32+ RGCs), p = 0.0422 (Other RGCs). ANOVA followed by Bonferroni test, two-sided. **B** Fluorescence images showing PRV-traced cells (green) co-localized with ipRGC marker Melanopsin (red) (left panel) and αRGC marker SMI32 (red) (right panel) in the 6 m Regen. group. Scale bar: 50 μm. **C** Schematic diagram illustrating the experimental design for (**D**–**F**). **D** Quantification of percentages of pupil constrictions during PLR tests in different groups at different times post-injury. Data presents mean ± SEM. n = 5 (Injury Ctrl.) and n = 6 (ipRGC Regen.). Statistical significance: p ≤ 0.0001, ANOVA

followed by Bonferroni test, two-sided. **E** Fluorescence images showing CTB-labeled RGC axons (green) and GFAP-labeled lesion site (red) in LGN and OPN from different groups. The dash lines outline the OPN regions. Scale bar: 200 μm. **F** Quantification of CTB-FITC intensities from labeled RGC axons at different distances from the lesion site. Blue asterisks indicate the comparison between 6 m ipRGC Regen. group and 12 m Injury Ctrl. group, while red asterisks represent the comparison between the 12 m ipRGC Regen. group and 12 m Injury Ctrl. group. Data presents mean ± SEM. n = 5 (12 m Injury Ctrl. and 6 m ipRGC Regen. group) and n = 6 (12 m ipRGC Regen.). Statistical significance: Red asterisks: p < 0.0001 (100 μm), p = 0.0412 (300 μm), Blue asterisks: p < 0.0001 (100 μm), p = 0.0034 (200 μm). ANOVA followed by Dunnett's test, two-sided. ns, not significant, p > 0.05, * p ≤ 0.05, ** p ≤ 0.01, *** p ≤ 0.001 Source data are provided as a Source Data file.

Melanopsin overexpression has been shown to increase light-induced RGC activity and thus can enable persistent and strong input[26,27]. After OTI, the survival of M1-M3 ipRGCs and alpha RGCs was not affected (Supplementary Fig. 14A). Melanopsin was overexpressed by delivering AAV-Melanopsin intravitreously to the mice, and approximately 80% of RGCs were melanopsin positive, compared to only 5% in the control mice (Supplementary Fig. 14B, C). The overexpressed melanopsin can be detected in non-ipRGCs, such as T-RGCs (Supplementary Fig. 14D). Interestingly, melanopsin overexpression in intact mice amplified PLR following 1 min 0.5 mW/cm$^2$ and 0.01 mW/cm$^2$ light stimulation, particularly under the lower light intensities (Supplementary Fig. 14E–H). AAV-Melanopsin was then intravitreally injected into PSCL Regen. mice 2 months after pre-OPN injury (PSCL+ Mela. OE), while another group received AAV-GFP injection as a control (PSCL + GFP) (Fig. 6A). Notably, in the PSCL Regen. mice, melanopsin overexpression had a higher PLR response at 3 months post-OTI than the control group (Fig. 6B). RGC axons in both groups regenerated and reinnervated OPN at similar levels (Supplementary Fig. 14I, J), excluding the possibility that stronger PLR induced by melanopsin overexpression stemmed from enhanced RGC axon regeneration. These results show that increasing RGC photosensitivity can further enhance PLR recovery after partial RGC axon regeneration post-OTI.

### RNA-Seq reveals downregulation of presynaptic release machinery and modulating presynaptic calcium channels improves PLR function

To further search for factors able to promote functional recovery, we performed single-cell RNA sequencing of retrogradely-labeled RGCs to examine the transcriptome dynamics during functional recovery. We manually isolated CTB-555-labeled OPN-projecting RGCs from PSCL Regen. mice and injury Ctrl. mice at differential time points (Supplementary Fig. 15A, B). Cells with high RGC marker gene expression were chosen for full-length mRNA library preparation and sequencing, and their identities were verified by integrating our scRNA-seq data with a published RGC atlas[65] (Supplementary Fig. 15C, D). Stringent filtering for sequencing quality and cell type identity left only around 10 RGCs in each PSCL group at each time point, making it statistically challenging to probe differences among these groups. Thus, we compiled all the data from PSCL group or Ctrl. group, respectively, across different time points together as pseudo-bulks, and found that the PSCL group was indeed enriched in genes related to axon regeneration, including widely reported genes such as *Sox11*, *Gal*, *Atf3*, and *Sprr1a*[66–69], as well as candidates such as *Serpina3n*, *Wfs1*, *Ecel1*, and *Parp10* based on reported associations with axon growth[52,70–73] (Supplementary Fig. 15E). Gene ontology analysis of differentially expressed genes indicated that the PSCL group upregulated genes associated with immune function and downregulated genes associated with synaptic function (Supplementary Fig. 15F, G). This observation is especially interesting because optimal functional recovery may require the regenerated axon to recapitulate a physiological state. We also checked the expression of genes known to be crucially involved in the presynaptic release machinery and found that *Vamp2*, *Snap25*, and *Syt1* were

downregulated during the regeneration and reconnection process (Fig. 6C), suggesting a potential functional compromise. Inspired by these results, we hypothesized that enhancing presynaptic release of the regenerated axons could promote functional axonal recovery. R-roscovitine, but not its stereoisomer S-roscovitine, has been demonstrated to increase Ca$^{2+}$ influx and transmitter release at presynaptic terminals by slowing the deactivation of voltage-gated calcium channels, although both stereoisomers are also Cyclin-dependent kinase 5 (CDK5) inhibitors[28–30]. To test our hypothesis, we systemically delivered R-roscovitine or S-roscovitine as a control (5 mg/kg, intraperitoneal) to animals with initial PLR recovery mediated by optic tract regeneration (Fig. 6D). R-roscovitine, but not S-roscovitine, enhanced PLR in regeneration mice from 17% to 29% (Fig. 6E and supplementary movie 4), demonstrating that indeed enhancing presynaptic release can further improve PLR recovery after some extent of post-injury axon regeneration and reconnection.

## Discussion

We developed an intracranial injury model for investigating functional axon rewiring, referred to as the pre-OPN OTI model. This model is operationally simple, utilizes well-established neuronal circuitry, and enables straightforward confirmation of complete injury as well as quantitative functional assays. The combination of Pten/Socs3 knockout with CNTF expression promotes axonal regeneration into the original target, the OPN, where new functional synapses are formed by the ipRGCs, partially restoring the PLR. However, the model does have certain limitations, such as the requirement for bilateral rather than unilateral axotomy due to the bilateral projection of RGC axons to the OPN, as well as a relatively slow recovery process. Compared to ONC, axons grow substantially slower after OTI. RGC axons take 6 months to reinnervate the OPN for ~1 mm after Pten/Socs3 knockout and CNTF expression modification, but intraorbital and pre-chiasm ONC under same manipulation allow RGC axons to grow 4–5 mm or 2–3 mm to the optic chiasm or hypothalamus, respectively, within 1 month[9,23]. The slower axon growth rate observed in the OTI model can be attributed to a variety of factors, including the distinct environments present in brain parenchyma after injury, the complex regrowth directions in the brain, the longer distance required for material transport from the soma to the axon, and the intact branches before the lesion site[24,40,74,75]. Those factors suppress the regeneration speed of the lesioned optical tract. Nevertheless, combining Lipin1 knockdown with Pten/Socs3 knockout and CNTF expression significantly shortens the recovery time from 6 months to 3 months. Interestingly, as an indicator of the retrograde axotomy signal, DLK pathway[41] activation in RGCs is delayed and weaker following OTI compared to ONC. This difference is partially explained by the fact that the OTI model leaves approximately 10% of RGCs uninjured. More importantly, the varying distances of axonal injury from the cell body and the presence of intact branches likely play a more significant role in shaping the distinct signaling responses[18,38–40]. Given that DLK signaling is also integral to the regeneration process, it warrants a study on whether activating DLK or other similar injury signaling without

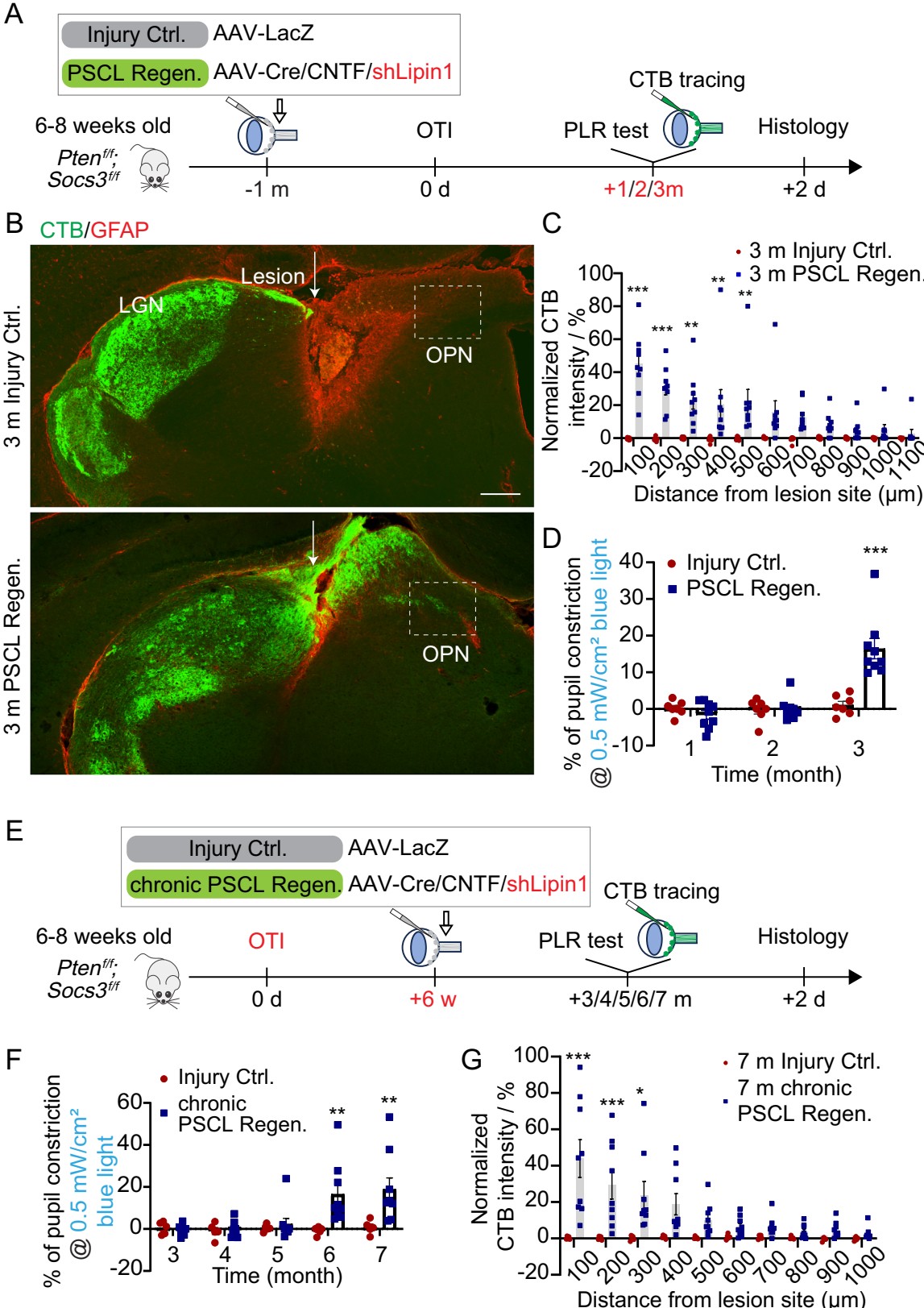

killing regenerating neurons may further facilitate the growth and recovery process.

After CNS injury, precise pathfinding of regenerated axons to reach their targets is vital for functional recovery[6,8,76]. This process may be affected by axon guidance molecules that steer axons in early brain development but differ in type, amount, and distribution in adult

CNS[24,77−82]. In our pre-OPN OTI model, axons successfully reconnected with their original targets in both the OPN shell and core, contrasting with the pre-chiasm or intraorbital ONC models where axons reached the hypothalamus but avoided the core of the SCN[9,83]. This indicates that the local cues in the SCN and OPN are distinct−SCN repels while OPN attracts regenerating axons. To explore how regenerated axons

**Fig. 5 | The combination of Lipin1 knockdown with Pten/Socs3 knockout and CNTF expression accelerates axon regeneration and functional recovery after pre-OPN OTI. A** Schematic diagram illustrating the experimental design for the combination of Lipin1 knockdown with Pten/Socs3 knockout and CNTF expression in the pre-OPN model. **B** Fluorescence images showing CTB-labeled RGC axons (green) and GFAP-labeled lesion site (red) in LGN and OPN from different groups. The dash lines outline the OPN regions. Scale bar: 200 μm. **C** Quantification of CTB-FITC intensities of labeled RGC axons at different distances from the lesion site. Data presents mean ± SEM. n = 7 (3 m Injury Ctrl.) and n = 9 (3 m PSCL Regen.) mice per group. Statistical significance: p < 0.0001 (100–200 μm), p = 0.0023, 0.0064, 0.0032 (300–500 μm). ANOVA followed by Bonferroni test, two-sided. **D** Quantification of percentages of pupil constrictions during PLR tests in different groups at different time points post-injury. Data presents mean ± SEM. n = 7 (3 m Injury Ctrl.) and n = 9 (3 m PSCL Regen.) mice per group. Statistical significance: p ≤ 0.0001, ANOVA followed by Bonferroni test, two-sided. **E** Schematic diagram showing the experimental design for chronic optic tract injury. **F** Quantification of percentages of pupil constrictions during PLR tests in different groups at different time points post-injury. Data presents mean ± SEM. n = 5 (Injury Ctrl.) and n = 9 (chronic PSCL Regen.). p = 0.0042, 0.0026 (6 m, 7 m). ANOVA followed by Bonferroni test, two-sided. **G** Quantification of the CTB-FITC intensities of the RGC axons at varying distances from the lesion site. Data presents mean ± SEM. n = 5 (7 m Injury Ctrl.) and n = 9 (7 m chronic PSCL Regen.). Statistical significance: p < 0.0001 (100 μm), p = 0.0009 (200 μm), p = 0.013 (300 μm). ANOVA followed by Bonferroni test, two-sided. * p ≤ 0.05, ** p ≤ 0.01, *** p ≤ 0.001 Source data are provided as a Source Data file.

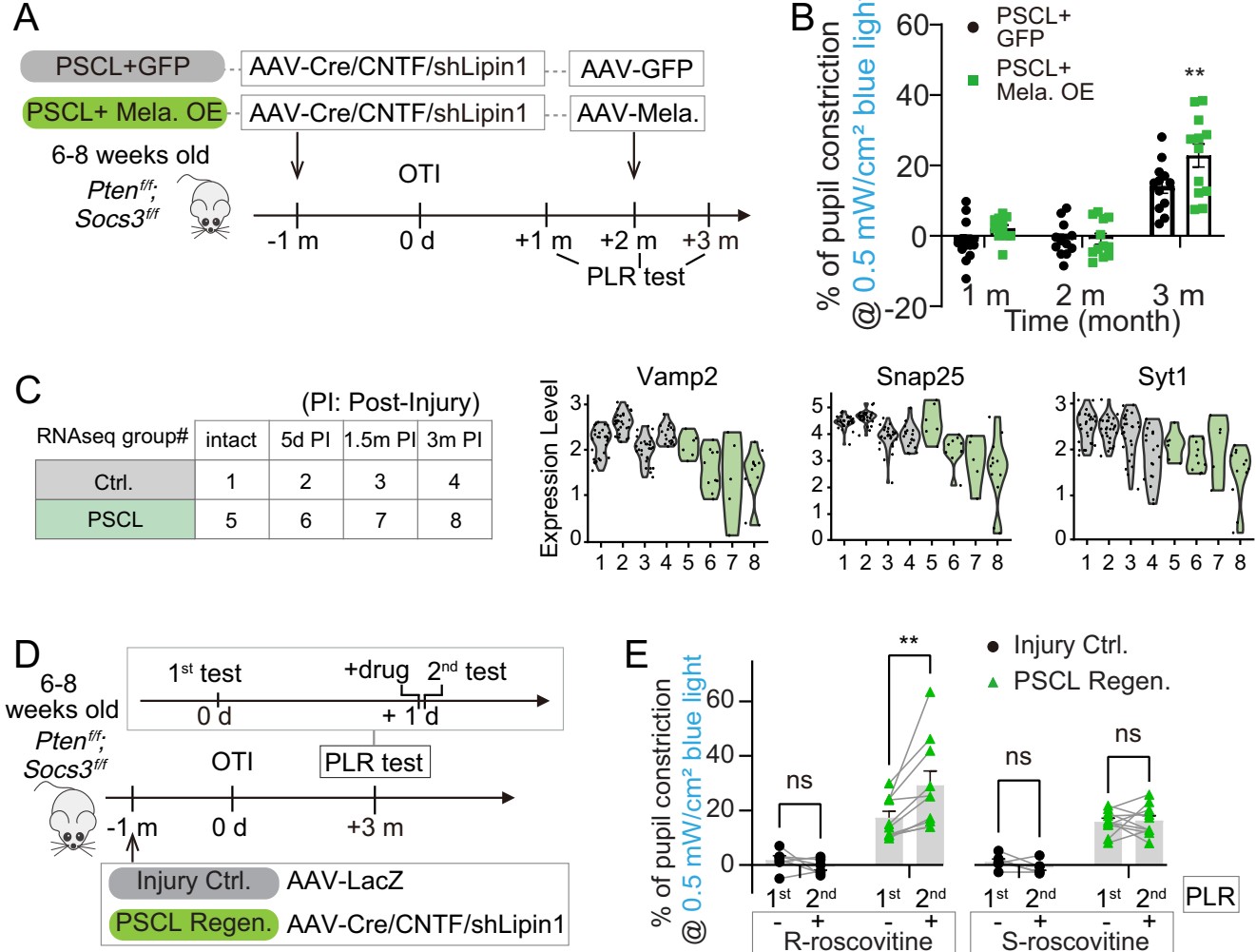

**Fig. 6 | Neuronal and presynaptic modulation improves functional recovery. A** Schematic diagram showing the experimental design for PLR tests in PSCL Regen. mice with or without melanopsin overexpression (OE). **B** Quantification of percentages of pupil constrictions during PLR tests in different groups at different time points post-OTI. Data presents mean ± SEM. n = 12 mice. Statistical significance: p = 0.0076 (3 m), ANOVA followed by Bonferroni test, two-sided. **C** RNAseq results reveal the gradually decreasing expression of several key genes involved in pre-synaptic release machinery in mice receiving AAV-Cre+AAV-CNTF + AAV-shLipin1 (PSCL) over time after injury, compared to the control group receiving only AAV-GFP injection (Ctrl.). **D** Schematic diagram showing the experimental design for PLR tests with or without R-roscovitine and S-roscovitine intraperitoneal administration at 3 months post-injury. Two PLR tests in response to 0.5 mW/cm² blue light stimulation were conducted with a 1-day interval. Twenty minutes prior to the 2nd test, mice received intraperitoneal injections of either R-roscovitine or S-roscovitine. **E** Quantification of percentages of pupil constrictions in different groups with 0.5 mW/cm² light stimulation. Data presents mean ± SEM. n = 6 (Injury Ctrl.) and n = 10 (PSCL Regen.). Statistical significance: R-roscovitine: p > 0.999 (Injury Ctrl.), p = 0.0062 (PSCL Regen.), S-roscovitine: p > 0.999 (Injury Ctrl., and PSCL Regen.), ANOVA followed with Bonferroni test, two-sided. ns, not significant, p > 0.05, ** p ≤ 0.01. Source data are provided as a Source Data file.

regrow along the correct trajectory to their original target, we combined OPN proteomics data with RGC single-cell RNA sequencing data collected 3 months after injury. OPN proteomics was carried out using mass spectrometry (MS) analysis of OPN regions. The analysis of both

data sets revealed the expression of axon guidance-related molecules in PSCL mice. The majority of the identified guidance receptors are from the immunoglobulin superfamily of cell adhesion molecules, which includes *Cadm2*, *Cntn1*, *L1cam*, *Ncam1*, *Ptprs*, and *Thy1*

(Supplementary Fig. 15H). The roles of these molecules in post-injury axonal regeneration and reconnection, as well as functional recovery, require additional experimental investigation. Nonetheless, these results demonstrate the versatile utility of the pre-OPN OTI model, enabling a comprehensive investigation of post-injury axon reconnections from multiple perspectives.

We found that most synapses mediating the PLR response were reformed by ipRGC axons after injury, as examined by PRV trans-synaptic labeling. We show quantitatively that ~30% of the typical pupil constriction can be restored by ~12% reconnected RGCs, most of which are ipRGCs. This indicates that regenerating ipRGCs can still recognize and reconnect with their original target neurons—a capability also critical for synaptic connections during development[34,84,85]. Moreover, stimulating axon regeneration in ipRGCs was sufficient for partial functional recovery. This result aligns with a recent study where directing spinal cord neuronal projection of a certain subtype to lumbar targets improved locomotion in a spinal cord crush model[8], suggesting that correct synapse formation between specific neuronal subtypes is a promising strategy for functional restoration in injured nervous systems. However, we did not specifically investigate which RGC subtypes regenerated better during the axon regeneration process, though we anticipate that many other RGC subtypes, in addition to M1-M2 ipRGCs and alpha-RGCs, could regenerate their axons across the lesion site according to a study using a similar strategy to promote axon regeneration in ONC model[67].

Neurons switch from a physiological state suitable for long-distance axonal projection to another state for local functional circuit integration once they reach the right targets, resulting in relatively stable circuitry[10,86–89]. We used knockout or knockdown approaches to reactivate neuronal intrinsic growth programs and promote axon regeneration without shutting them off. It is possible that those regenerated neurons do not fully transition from the "growth state" to the "functional state" after reaching their destinations, whereas dorsal root ganglion neurons revert to their pre-injury state once functional reconnection is established following sciatic nerve crush[90,91]. Indeed, we found the synaptic function genes, including presynaptic release machinery, were downregulated in the regeneration group. In addition, the pre-OPN OTI model transitions from a complete injury model to an incomplete injury model after some axons reconnect with their original targets. This shift enables the exploration of mechanisms that might enhance functional recovery at a given degree of axon regeneration. Based on these rationales, we identified two factors—increased RGC photosensitivity and enhanced presynaptic release—that can augment PLR recovery following partial axon regeneration, providing therapeutic options with potential applications in other injury models. Collectively, these findings highlight the applicability of our pre-OPN OTI model in studying not only axon regeneration and reconnection after complete injuries but also mechanisms and approaches to further improve functional recovery under the settings of incomplete injuries and across differential CNS injury models.

## Methods

### Animals

All experimental procedures were conducted in accordance with animal protocols (A18042, AEP-2022-0083, AEP-2023-0069) approved by the Laboratory Animal Facility at the Hong Kong University of Science and Technology. *C57BL/6 J* mice were obtained from Charles River. The *Pten^{f/f};Socs3^{f/f}* double-floxed mice[23] were provided as a gift by Prof. Zhigang He (Boston Children's Hospital). *Opn4^{Cre}*[92] mice were obtained from Prof. Samer Hattar at Johns Hopkins University. The *Opn4^{Cre}* mice were crossed with *Pten^{f/f};Socs3^{f/f}* mice to generate *Opn4^{Cre};Pten^{f/f};Socs3^{f/f}* mice. *Opn4-GFP* mice were obtained from the Mutant Mouse Regional Resource Center, an NIH funded strain repository. *Vglut2-Cre*[93] mice (stock#: 016963) were purchased from Jackson Laboratory. 6-8 weeks old mice of both genders were used in this study. The animals were

habituated in a controlled environment with a 12:12-hour light-dark cycle, a temperature maintained at 20–24 °C, relative humidity ranging from 30–70%, and provided with ad libitum access to food and water.

### AAVs packaging

AAV serotype 2/1 was used for the expression of Cre. AAV serotype 2/2 was used for the expression of LacZ (Vector Biolabs Cat No. 7003), GFP (Addgene plasmid# 105539), DIO-GFP (Addgene Plasmid# 51504), Cre (Penn Vector, Cat# P1848), CNTF (same as one in previously published work[6]), dAPEX2 (Addgene plasmid# 117176), hM3D(Gq) (Addgene plasmid# 50463), Melanopsin (same as the one in our previous work[26]), and shRNA (Control-shRNA: GACCATCAATATGACTAGA; Lipin1-shRNA: CGTGTCATATCAGCAATTT, the efficacy of Lipin1 knockdown was demonstrated in our previous work[54,56]). AAV serotype 2/9 was used for the expression of DIO-EGFP-TVA (BrainVTA, Cat# PT-0021), DIO-RVG (BrainVTA, Cat# PT-0023), and RV-ENVA-ΔG-dsRed (BrainVTA, Cat# R01002). The virus titer was measured using qRT-PCR, with the typical titer used being $10^{13}$ genome copies per milliliter.

### Pre-OPN optic tract injury model

The 6-8 weeks old *Pten^{f/f};Socs3^{f/f}* mice were anesthetized with ketamine (80 mg/kg) and xylazine (10 mg/kg). This anesthesia protocol was consistently applied for all subsequent surgeries. AAV-Cre and AAV-CNTF (2 μl) were then intravitreally injected into the left eye of the mouse, while AAV-LacZ was used as control. Four weeks after virus delivery, we conducted a pre-OPN lesion surgery. After the mice were anesthetized, the scalp was cleaned with iodine, and an incision was created to expose the skull. Symmetrical rectangular cranial windows were drilled, and the bone flap was removed to expose cortical tissue. The bilateral optic tracts were completely crushed by forceps (#5 Domont, 11295-10, Fine Science Tools) between the LGN and OPN/SC, the lesion site was situated adjacent to the LGN. Before the surgery, the forceps were carefully checked to ensure that the smooth inner surfaces at the tips could close flat to 5 mm under pressure. During the surgery, the forceps were fixed to the stereotaxic holder with a 3 mm distance between the two tips. The left tip of the forceps was gently placed on the top of the bregma, which was set as the zero point. The left tip was then moved to the coordinates (anteroposterior −3.6 mm, mediolateral ±1.7 mm), while the right tip was positioned at anteroposterior −0.6 mm and mediolateral ±1.7 mm. Bilateral optic tract crush was then performed at a depth of 3.5 mm for 5 seconds. The blood was meticulously cleaned, the bones were returned, and the incision was sutured. Meloxicam (1 mg/kg) was subcutaneously injected for analgesia. A heating pad was used to maintain the body temperature of the mice during the surgery until they were fully recovered. The protocols for analgesia and maintaining the mice's body temperature were consistently applied for all subsequent surgeries.

The success of the surgery was confirmed using a two-color CTB injection to assess whether any axons were spared beyond the lesion site. First, CTB-FITC (Sigma-Aldrich, Cat# C1655) was injected intravitreously in both eyes, followed by bilateral OTI surgery two days later. One day after the OTI, CTB-555 (Invitrogen, Cat# C34776) was injected intravitreously in both eyes, and the mice were sacrificed two days later. The two-color CTB labeling was analyzed in the OPN, mdPPN, NOT, and SC, which are located behind the lesion site. The absence of CTB-555 signaling in these nuclei, which had been pre-labeled with CTB-FITC, indicated that the lesion was complete in this surgical model. Mice with LGN damage from the OTI surgery were excluded from further quantification and analysis. The PLR test was also performed on every mouse after injury, and only the mice whose pupils did not respond to light stimulation after pre-OPN OTI indicated the completion of the surgery. The histological and behavioral evaluations indicated that the overall success rates of our OTI model exceed 89%. To examine axon regeneration at various time points after pre-OPN

OTI, the control and regeneration mice were sacrificed 2 days after CTB tracing.

In the chronic pre-OPN model, AAV-Cre, AAV-CNTF, and AAV-Lipin1-shRNA were intravitreally injected 6 weeks after pre-OPN lesion, and PLR was tested after AAV injection at different injury stages.

## Optic nerve crush
Optic nerve crush was conducted in accordance with previously reported[11]. After the mice were anesthetized, the conjunctiva of the eye was gently exposed by clamping to the eyelid. An incision was made on the conjunctiva to expose the optic nerve carefully. The optic nerve was then crushed using forceps (Dumont #5, Fine Science Tools) for 2 s, approximately 1–2 mm distal to the optic disk.

## Anterograde tracing of RGC axons
For the tracing of RGC projected axons, the CTB-FITC 1 μg/μl was administered via intravitreal injection into the vitreous body of the mouse. After the mice were anesthetized, the eyelid was gently clamped to expose the conjunctiva, and a Hamilton syringe was used to extract 2 microliters of vitreous body from the eye. Then, 2 μL of CTB was carefully injected into the vitreous body. Eye ointment was applied after the procedure to protect the eye.

## Immunohistochemistry
The samples were incubated with the appropriate antibodies as required for the experiment. Primary antibodies: Rabbit anti-FITC (Invitrogen, 71-1900), Mouse or Rabbit anti-Tuj1 (BioLegend, 801202 and 802001), Mouse anti-SMI32 (BioLegend, 801701), Rabbit anti-Melanopsin (Advanced Targeting System, AB-N39), Mouse anti-GFAP (BioLegend, 808402), Rabbit anti-c-Fos (Cell Signaling Technology, 2250), Chicken anti-GFP (Invitrogen, A10262), Rabbit anti-Homer1 (Millipore, ABN37), Mouse anti-Bassoon (Abcam, ab82958), Chicken-anti-Tbr1 (Millipore, AB2261), Chicken anti-Tbr2 (Millipore, AB15894), Mouse anti-MBP (BioLegend, 808402), Rabbit anti-Phospho-c-Jun (Cell Signaling Technology, 3270), Mouse anti-Phospho-c-Jun (Invitrogen, MA5-27760), Mouse anti-parvalbumin (Millipore Sigma, P3088). Secondary antibodies: Goat anti-Mouse 555 (Invitrogen, A-21424), Goat anti-Mouse Cy5 (Invitrogen, A-10524), Goat anti-Rabbit 555 (Invitrogen, A-21429), Goat anti-Rabbit Cy5 (Invitrogen, A-10523), Goat anti-Chicken 647 (Invitrogen, A-21449) Goat anti-biotin (Thermo Fisher Scientific, B-2770).

Mice were anesthetized with a lethal dose of anesthesia and perfused transcardially with PBS followed by 4% paraformaldehyde (PFA, Sigma-Aldrich, 30525-89-4). The retinas and brains were dissected out and postfixed in PFA overnight at 4 °C. Mice brains were dehydrated in 15% sucrose (Invitrogen, 15503022) followed by 30% sucrose at 4 °C and then embedded in OCT compound (SAKURA, 4583) using dry ice before sectioned into 40 μm slices (25 μm for retina, 8 μm for optic nerve). For the staining, slides were blocked and permeabilized with 4% normal goat serum (NGS, Invitrogen, 50062Z) containing 0.1% TritonX-100 (Sigma-Aldrich, T8787) for 30 min. The slides were then incubated with primary antibody (1:500) overnight, washed 3 times with PBS, and incubated with secondary antibodies (1:500) for 2 h. After three additional PBS washes, the slides were mounted with coverslips and imaged using confocal microscopy (Zeiss, LSM 880). To amplify CTB signals of traced axons, Elit ABC Kit (vector laboratories), and tyramide signal amplification (TSA) Cyanine 5 system was used.

## Quantification of axon regeneration
The intensity of CTB-labeled regenerating axons was measured using ImageJ at different distances from the lesion site, with the lesion site being the zero point. A rectangle with dimensions of 100 μm × 300 μm was drawn to encompass the regenerated axons every 100 μm from the lesion site to the OPN region. The rectangle's height (300 μm) was chosen to cover most of the area with regenerated axons. The intensity of CTB in the regenerated axons was assessed by comparing it to the axons before the lesion site (defined as 100%), which was measured 100 μm rostral to the lesion site. The average percentage of labeled CTB intensity was then calculated for each animal by averaging the results from various brain coronal sections containing the OPN region.

## Quantification of RGC survival
To determine the number of surviving RGCs, whole-mount Tuj1 staining was performed. The retina was carefully exposed and dissected after fixation as previously mentioned. After incubation in 4% NGS for 30 min in a 24-well plate, the retina was incubated with Tuj1 antibody (1:1000) overnight. Following washes with PBS, the retina was incubated with a secondary antibody (1:500) for 2 h at room temperature. After another wash with PBS, the retina was mounted onto glass slides, and images were captured from both the peripheral and central regions of each retina using a confocal microscope.

## Quantification of axon terminal boutons and synapses
Two days after CTB-FITC intravitreal injection into the eyes of Sham, 6 m Injury Ctrl., and 6 m Regen. mice, the OPN sections were stained with FITC antibody for axon terminal bouton analysis, or triple-stained with FITC, Bassoon, and Homer1 antibodies for synapse analysis. Super-resolution images were acquired using the ZEISS Elyra 7 microscope, and the number of axon terminal boutons and synapses was quantified using ImageJ. The percentage of synapses in CTB-labeled boutons was calculated by dividing the number of synaptic boutons that were double-positive for FITC and Bassoon and adjacent to Homer1 by the total number of CTB-labeled RGC axon terminal boutons in the OPN. It is important to note that the Homer1-stained postsynaptic structures may be localized in other layers that are not detectable by microscope scanning, which is why the percentage of synapses in CTB-labeled boutons in the Sham mice does not reach 100%.

## Pupillary light reflex test
Before PLR recording, the mice underwent a 2 h period of dark adaptation. Subsequently, the mice were lightly anesthetized and restrained on an appropriate mouse holder. During PLR recording, a monochromatic light (480 nm) was used to stimulate the mouse's eye for one minute. The stimulator device was designed to cover the eye completely, preventing light from affecting the contralateral eye. An infrared camera was positioned opposite to the stimulated eye to record the consensual PLR. After recording, the area of the pupils before and after light stimulation was measured by using ImageJ (https://imagej.net/ij/, RRID: SCR_002285), and quantitative analysis was performed by comparing the reduced pupil area (pupil area before light stimulation minus pupil area after light stimulation) with the pupil area before light stimulation.

For the mice that underwent the PLR test after R-roscovitine (MedChemExpress, HY-30237) and S-roscovitine (AdipoGen, AG-MR-C0002-M025) administration, they were dark-adapted for 2 h before receiving an intraperitoneal injection of R-roscovitine (5 mg/kg) or S-roscovitine (5 mg/kg). After 20 min, the PLR test was conducted with 0.5 mW/cm² blue light stimulation.

For the mice that underwent the PLR test under DREADD-Gq stimulation, they were dark-adapted for 2 h, and their pupils were recorded with an infrared camera in the dark. Then, CNO (5 mg/kg, DC chemicals, DC7991) was intraperitoneally injected, and 30 min later, their pupils were recorded in the dark again. The area of the pupils before and after the CNO injection was measured for analysis.

## In-vivo electrophysiology recording
Mice were typically anesthetized, and dexamethasone (2.5 mg/kg) was administered subcutaneously to prevent brain edema. The mice were positioned on a stereotaxic frame, with body temperature maintained

at 37 °C throughout the procedure. A rectangular window covering the brain region of OPN was opened in the right hemisphere, which was contralateral to the AAV-injected eye. The coordinates for OPN placement were anteroposterior: −2.5 mm, and mediolateral: 0.6 mm. A microelectrode array (with a 2 × 8 configuration, 35 μm diameter, made by Plexon Inc.) was positioned atop the OPN according to these coordinates and then inserted to a depth of 2.3 mm into the OPN. Blood was carefully cleared away with cold saline, and dental cement was used to secure the microelectrode. Signals were recorded 1-week post-surgery. During neural data collection, the extracellular potential signals were simultaneously recorded from the OPN region using the 16-channel microelectrode array, with data acquisition facilitated by a Plexon system (Plexon Inc., Dallas, Texas). The raw neural signals were sampled at 40 kHz and subjected to high-pass filtering at 500 Hz employing a 4-pole Butterworth filter. The spikes were detected from the processed neural signal by using a threshold at −4.5σ, where σ represents the standard deviation of the amplitudes. Afterward, the offline sorter (Plexon Inc., Dallas, Texas) was employed to sort single neurons from each channel. These sorted spikes were counted by using a non-overlapping window of 100 ms duration. Data analysis was specifically focused on a steady recording period (30 s before and after light stimulation) obtained during 15 cycles of alternating 60 s exposures to light (intensity of 0.5 mW/cm²) and darkness.

To confirm that the microelectrode was inserted into the correct location, AAV-hM3D(Gq) (300 nl) was injected into the OPN four weeks before the electrophysiology surgery. A PLR test was conducted to assess whether activation of OPN neurons by CNO administration could induce pupil constriction. During the electrophysiology recording, OPN neuronal firing rate induced by CNO (5 mg/kg, i.p.) was recorded before and 30 min following administration to confirm the successful microelectrode placement.

### Transmission electron microscopy (TEM)
AAV-Matrix-dAPEX2 was intravitreally injected into mice, which were sacrificed four weeks after virus injection. The mice received a lethal dose of anesthetic and were perfused transcardially with warm Ame's medium, followed by 2% paraformaldehyde (PFA)/2.5% glutaraldehyde (Sigma-Aldrich, 111-30-8) in cacodylate buffer (CA, Sigma-Aldrich, 6131-99-3). The brains were then dissected and post-fixed at 4 °C overnight. The following day, the brain was cut into 100 μm slides using a vibratome. Brain sections were treated with 50 mM glycine (Sigma-Aldrich, 56-40-6) for 10 min to quench residual aldehyde activity, followed by a wash with CA. Subsequently, the sections were incubated in 0.3 mg/ml 3′3-diaminobenzidine tetrahydrochloride hydrate (DAB, Sigma-Aldrich, 868272-85-9) for 30 min. After that, $H_2O_2$ was added to the DAB solution to achieve a 0.003% concentration, the sections were then incubated in this solution for another hour. Sections were washed with CA four times for 10 min each and post-fixed with 3% glutaraldehyde overnight at 4 °C. On the third day, brain sections were washed with CA for 10 min and incubated with 50 mM glycine for 10 min. The target brain regions were then dissected out and placed into 2% osmium tetroxide (Electron Microscopy Science, 19190) for an hour followed by 1 h in 2.5% potassium ferrocyanide (Sigma-Aldrich, P9387). After washing with $ddH_2O$ four times for 10 min each, the sections were treated with 1% thiocarbohydrazide (Sigma-Aldrich, 223220) for 15 min at 40 °C. The sections were washed with $ddH_2O$ four times for 10 min each and exposed to 2% osmium tetroxide in $ddH_2O$ for 1 h. The sections were then placed into 1% uranyl acetate for overnight staining at 4 °C. After washing the sections with $ddH_2O$ four times for 5 min each, they were dehydrated using a series of ethanol solutions (50%, 70%, 90%, and 95%) for 5 min each at 4 °C. The sections were then immersed in pure ethanol for two 20 min sessions at room temperature, followed by three 10 min sessions in acetone. Next, the sections were infiltrated with 25% resin in acetone for 2 h and placed into 50% resin overnight. The following day, the sections were transferred into 75%, 100% resin for 2 h each. Finally,

the sections were embedded in 100% resin and oven-cured at 60 °C for 72 h. The samples were then cut into 80 nm for imaging. Images were captured by Hitachi H-7650 transmission electron microscope operated at 80 kV.

### PSD length measurement
After capturing the EM images, the length of the PSD was measured manually using ImageJ. For quantification, only synapses with clearly defined structures were included. The PSD length was measured by manually drawing a line along the PSD in the postsynaptic membrane, utilizing the calibrated scale of the EM images[94].

### Neuronal activity training
To investigate the impact of neuronal activity on PLR recovery, we utilized two methods of neuronal activity training: light-dark stimulation training and DREADD-Gq + CNO training. Two months after optic tract injury in the PSCL manipulation mice, the mice were randomly divided into three groups. One group received an intravitreous injection of AAV-hM3D(Gq). At 3 months post-injury, the AAV-hM3D(Gq)-injected mice received DREADD-Gq + CNO training, while the remaining two groups received light-dark stimulation training and no training as control. For the light-dark stimulation training, the mice were moved to a cage with a blue light array attached outside the cage (light intensity in the cage: 0.6 mW/cm²), with the light turned ON and OFF in a 1 h interval repeatedly. For the DREADD-Gq + CNO training, the mice received intraperitoneal injections of CNO twice a day with a dosage of 5 mg/kg. The non-training group was kept in normal circadian conditions (12:12 light-dark cycle). The training started at 3 months post-injury and lasted for one month, with PLR recorded monthly before training and post-training.

### Increasing RGC photosensitivity
To investigate whether enhancing the photosensitivity of injured RGCs promoted PLR recovery, AAV-Melanopsin was intravitreally injected into PSCL manipulation mice 2 months after pre-OPN lesion, with AAV-GFP injection as a control. PLR was recorded 1 month before and after AAV injection.

### CTB Retrograde labeling
For CTB retrograde labeling, mice were anesthetized as previously mentioned, and secured on the stereotaxic apparatus. To inject CTB into the OPN, a small window was opened over the OPN (anteroposterior −2.5 mm, mediolateral ± 0.6 mm). Then, CTB555 (0.3 μl) was injected into OPN at a depth of 2.3 mm. To inject CTB into the SC, a cranial window was created with an anteroposterior range of approximately −3.4 mm to −4.1 mm and a mediolateral range of ±1 mm. Then, 0.2 μl of CTB555 was injected at the following coordinates: 1. anteroposterior −3.4 mm, mediolateral ±0.25 mm and ±0.75 mm, dorsoventral 1.5 mm and 1.625 mm corresponding to the respective mediolateral coordinates; 2. anteroposterior −3.64 mm, mediolateral ±0.25 mm, ±0.5 mm, ±0.75 mm and ±1 mm, dorsoventral 1.125 mm, 1.25 mm, 1.375 mm and 1.4 mm corresponding to the mediolateral coordinates respectively; 3. anteroposterior −4.04 mm, mediolateral ±0.25 mm, ±0.5 mm, ±0.75 mm and ±1 mm, dorsoventral 1.125 mm, 1.25 mm, 1.375 mm and 1.5 mm. Three days post-CTB tracing, RGCs were harvested for single-cell RNA sequencing or histological analysis.

### PRV-GFP multisynaptic tracing
To perform multisynaptic retrograde tracing, PRV-GFP (BrainVTA, Cat# P03001) was intracamerally injected into the eye without AAV injection (right eye). After the mice were anesthetized, a 32 G needle was used to create a hole (about 0.5 mm from the limbus) in the anterior chamber. After withdrawing the needle, wait for 10 min to allow the aqueous humor to release. Next, 1 μl of PRV-GFP was injected into the anterior chamber by the glass micropipette through the hole

carefully to avoid diffusion into the vitreous chamber. After injection, the glass micropipette was slowly withdrawn and veterinary antibiotic ointment was applied to the eye. After five days, the mice were sacrificed for further analysis. If any mouse from either group showed a large area of GFP-positive signals in the margin of the retina of the PRV-injected eye, this would indicate that the PRV had diffused into the vitreous, and the mouse would be excluded from the analysis.

## Rabies virus monosynaptic tracing

To perform monosynaptic retrograde tracing, a mixture of AAV9-Ef1α-DIO-His-EGFP-2a-TVA-WPRE-pA, AAV9-Ef1α-DIO-oRVG-WPRE-pA, AAV1-Syn-Cre, and glycoprotein-deleted rabies virus expressing dsRed (RV-ENVA-ΔG-dsRed) was injected into the OPN on the side opposite to the AAV injection eye. After anesthetizing the mice, a small window was made over the OPN, and 0.2 μl of the mixed virus (AAV9-DIO-EGFP-TVA, AAV9-DIO-RVG, and AAV1-Cre) was injected into the target OPN regions (anteroposterior −2.5 mm, mediolateral 0.6 mm, dorsoventral 2.3 mm). Three weeks later, 0.1 μl of RV-ENVA-ΔG-dsRed was injected into the same location. One week after this injection, the mice were sacrificed for further analysis.

## LPC optic tract injection

To investigate the impact of optic tract demyelination on PLR, mice were anesthetized and secured on the stereotaxic apparatus similar to the setup used for the retrograde labeling experiment. A small window was created above the optic tract with an anteroposterior distance of −2.5 mm, and 200 nl of 1% Lysophosphatidylcholine (LPC, Sigma-Aldrich, L4129) was injected at three mediolateral coordinates: 1.0 mm, 1.4 mm, and 1.8 mm, at depths of 2.2 mm, 2.3 mm, and 2.4 mm, respectively by the nanoject II (Drummond, Cat. Nos. 3-000-204) device. PLR measurements were taken one week after the LPC brain injection.

## RGC isolation and RNA sequencing

RGC isolation and RNA sequencing were performed as previously described[95]. The RGC-labeled retina was quickly dissected out from mice after cervical dislocation and incubated with 0.5 mg/ml papain (Sigma-Aldrich, P4762) at 37 °C for 30 min. The retina was then washed with Neurobasal-A medium and triturated into a single-cell suspension by pipetting. After centrifugation at 300 g for 6 min at 4 °C, the retinal cells were resuspended in Neurobasal-A medium supplemented with 10% B27.

## Bulk RNAseq

For bulk RNAseq, fluorescence-activated cell sorting (FACS) was used to isolate labeled RGCs. AAV-DIO-GFP was injected into the vitreous body of vGlut2-cre mice eye to label RGCs one month before any injury was applied to the animals. FACS gating was performed to isolate the desired population by sequentially excluding dead cells (DAPI-negative), selecting singlet cells, and then gating on GFP-positive cells to enrich the GFP-expressing population. 200 cells from one eye were collected into one tube containing smart-seq2 lysis buffer as a biological replicate. The RGC lysate underwent further processing according to the Smart-seq2 protocol. RNA was reversely transcribed into cDNA using superscript II (Invitrogen, 18064071) and TSO primer and followed by amplification with KAPA HiFi HotStart ReadyMix (Roche, KK2601) and ISPCR primer for 19 cycles. The amplified product was purified using Ampure XP beads. A fragment analyzer assessed cDNA quality and Qubit dsDNA High Sensitivity (HS) kit (Invitrogen, Q32854s) measured cDNA concentration. qPCR measured the relative expression level of the *Sncg* and *Rbpms* genes. Then the RGC cDNA was sent to BioCRF in HKUST for full Next Generation Sequencing (NGS) service. Raw reads were aligned to the GRCm38 (mm10) mouse reference genome using the "Subread" aligner from the Rsubread package, and gene counts were generated using the featureCounts function.

## Single-cell RNAseq

For single-cell RNAseq, mouth-pipetting was employed to pick single-labeled RGCs manually. A small volume of the RGC suspensions was transferred through a series of 1% BSA-containing PBS drops until a single targeted RGC was isolated. This RGC was then transferred into 4.2 μl lysis buffer, as per the Smart-seq2 protocol[96], and stored at −80 °C for subsequent processing.

The single-cell RGC lysate underwent further processing according to the Smart-seq2 protocol. RNA was reversely transcribed into cDNA using superscript II and TSO primer and followed by amplification with KAPA HiFi HotStart ReadyMix and ISPCR primer for 27 cycles. The amplified product was purified using Ampure XP beads. A fragment analyzer assessed cDNA quality and Qubit dsDNA High Sensitivity (HS) kit measured cDNA concentration. qPCR measured the relative expression level of the *Sncg* and *Rbpms* genes. Samples with low cDNA quality or low expression levels of both genes were excluded from downstream library construction.

Library construction utilized the TruePrep DNA Library Prep Kit V2 for Illumina (vazyme, TD501) and TruePrep Index Kit V2 for Illumina (vazyme, TD202), following the manufacturer's instructions. The constructed library was then sent to Novogene company for sequencing with NovaSeq PE150. Raw reads were aligned to the GRCm38 (mm10) mouse reference genome using the "Subread" aligner from the Rsubread package, and gene counts were generated using the featureCounts function[97].

## Bulk RNAseq analysis

A pre-filtering was applied to keep only genes that have a count of at least 10 in at least 3 samples. The DESeq2 package was used to test for differential expression among different groups[98]. PCA plot was produced by combining plotPCA function and ggplot function. To display the gene expression with scatterplot, we modified the vsScatterPlot function from the vidger package to label the DEGs overlapped with the DLK-dependent ONC-induced genes[41,99]. Heatmap plot was produced by using pheatmap function by supplying the gene list either from a specific gene ontology term or a manually curated list.

## Single-cell RNAseq analysis

Single-cell RNAseq analysis was conducted mainly using the "Seurat" package[100,101]. Cells with fewer than 200 detected genes and genes expressed in fewer than 3 cells were excluded from downstream analysis. To verify the identity of the cells, we integrated data from our cell population with a published RGC dataset using the FIRM package[102] and performed dimensional reduction analysis using the RunPCA and RunUMAP functions[65]. The integration UMAP plot was generated with DimPlot function to examine the distribution of our cells. Cells from our dataset that fell into the clusters annotated as RGCs in the referenced dataset were considered validated RGCs; some cells formed a single isolated cluster that did not resemble any RGC subtypes previously identified in the published atlas were considered as non-RGCs, and these cells were not used in the subsequent RGC analysis. This classification was further confirmed by checking for specific retinal cell type marker expression using DotPlot. Normalization, feature selection, and scaling of the filtered data were performed using NormalizeData, FindVariableFeatures, and ScaleData functions. After this stringent filtering, there were a limited number of RGCs remaining in PSCL subgroups, therefore we performed differential expression analysis between the whole GFP group and the whole PSCL group with FindMarkers. The differentially expressed genes (DEGs) were plotted in a volcano plot by EnhancedVolcano function with annotation to interesting genes. Gene ontology analysis of the DEGs was carried out with the "clusterProfiler" package to identify enriched pathways, and dotplot was used to show the enriched pathways. Expression of specific genes involved in interested pathways was plotted using VlnPlot from the "Seurat" package. To identify potential guidance-related

molecules, we filtered a manually curated list of axon guidance-related ligand-receptor genes based on their expression levels in the PSCL-3mpi group, using a threshold of 0.5 to define the significant expression.

## Protein extraction

The OPN region was isolated from the mouse brain and homogenized in RIPA buffer (50 mM Tris/HCl, 150 mM NaCl, 1% NP40, 0.25% sodium deoxycholate, 1 mM EDTA, pH 7.4) supplemented with protease inhibitor cocktails. The tissue lysate was rotated at 4 °C for 30 min, followed by centrifugation at $10,000 \times g$ for 15 min. The resulting supernatant was diluted in pre-chilled acetone at a ratio of 1:5. The samples were submitted to BioCRF at the Hong Kong University of Science and Technology (HKUST) for mass spectrometry analysis using a Bruker timsTOF Pro Mass Spectrometer.

## Proteomics data analysis

The proteomics pipeline FragPipe, developed by Prof. Alexey Nesvizhskii's group, was utilized for the analysis of mass spectrometry-based proteomics data, encompassing MSFragger, Philosopher, IonQuant, and other toolkits[103–105]. Raw timsTOF PASEF data were processed through FragPipe for peptide identification, FDR filtering, and label-free quantification. The "MSstats" package facilitated the statistical analysis of proteomics data output from FragPipe[106]. To identify potential guidance-related molecules, we filtered a manually curated list of axon guidance-related ligand-receptor genes based on their expression levels in the OPN-PSCL-3mpi group, with a stringent criterion of protein detection in at least 3 out of 4 animals to ensure robustness and reproducibility.

## Statistical analyses

GraphPad Prism 9 software was used for statistical analyses. The comparison between two groups was conducted using the student's t-test, and comparisons among multiple groups were analyzed by ANOVA. Variation within groups was represented by the standard error of mean (SEM). Significance was denoted as * $p \le 0.05$, ** $p \le 0.01$, *** $p \le 0.001$.

## Reporting summary

Further information on research design is available in the Nature Portfolio Reporting Summary linked to this article.

## Data availability

The RNA-seq data generated in this study are publicly available through GEO repositories (NCBI GEO: GSE267938). The mass spectrometry proteomics data have been deposited to the ProteomeXchange Consortium (https://proteomecentral.proteomexchange.org) via the iProX partner repository[107,108] with the dataset identifier PXD052573. The RNA-seq data utilized but not generated in this study are available at GEO repositories (NCBI GEO: GSE137400[65]). Any additional information required to reanalyze the data reported in this paper is available from the lead contact upon request. Source data are provided with this paper.

## Code availability

Custom code required to reanalyze the data reported in this paper is available from the lead contact upon request.

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

## Acknowledgements

This study was supported by grants from the Hong Kong Research Grant Council (AoE/M-604/16, C6034-21G, T13-602/21N, 16102524, JLFS/M-604/24 to K.L., C4002-21EF, C4014-23G, CRS_CUHK405/23 to L.J., C4001-22Y to L.D., and PDFS2223-6S04 to C.Y.), Innovation and Technology Commission (ITCPD/17-9), Hong Kong Center for Neurodegenerative Diseases InnoHK of Hong Kong SAR, National Natural Science Foundation of China (82171384), Guangzhou Key Projects of Brain Science and Brain-Like Intelligence Technology (20200730009), Shenzhen-Hong Kong Institute of Brain Science-Shenzhen Fundamental Research Institutions (2019SHIBS0001), Nan Fung Life Sciences to KL.

## Author contributions

Xin Z. performed optic tract injury, PLR test, immunohistochemistry, neuronal activity training, CTB retrograde labeling. Chao Y. injected virus. Xin Z., Chao Y., and Y.M. set up the PLR test system. C.Z. and S.T. conducted RNA-Seq study and data analysis. Xin Z., J.G., and LWJ contributed to the EM imaging. L.C. took super-resolution images. Xin Z., Xiang Z., S.C., Y.W., and W.H.Y. contributed to electrophysiology studies. Y.Z. injected PRV for retrograde labeling. X.W. packaged AAVs. J.Y.C. did the PLR video editing. K.L., L.D., Xin Z., Chao Y., C.Z., and J.W. designed the project, analyzed the data, and wrote the paper. All authors helped to prepare or edit the manuscript.

## Competing interests

The authors declare no competing interests.
