## [Transparent Peer Review file · Nature Communications]

Functional Optic Tract Rewiring via Subtype- and Target-specific Axonal Regeneration and Presynaptic Activity Enhancement

Corresponding Author: Professor Kai Liu

This manuscript has been previously reviewed at another journal that is not operating a transparent peer review scheme. The manuscript was considered suitable for publication without further review at Nature Communications.

Version 0:

Reviewer comments:

Reviewer #1

(Remarks to the Author)

In this manuscript, Zhang et al report a series of compelling results suggesting that regenerated axons induced by a set of combinatorial treatments (in elevating the intrinsic regenerative ability of adult neurons) could reinnervate physiological targets, make functional synapses and mediate partial behavioral recovery in a nicely designed optic tract injury model. Although previous studies have demonstrated the anatomical regeneration by these treatments, the resultant functional recovery have been difficult to assess. Here the authors first describe an intracranial pre-olivary pretectal nucleus (OPN) injury model. With this, they showed that regenerating axons induced by co-deletion of PTEN/SOCS3 and administration of CNTF are able to reach OPN. By both EM and electrophysiological means, they convincingly show the synapses are formed by these regenerating axons. Importantly, they also showed a partial recovery of pupillary light reflex. Additional over-expression of Lipin1 was able to facilitate axon regeneration and also accelerates behavioral recovery. Importantly, they also demonstrated that these treatments are effective in a chronic model. To further enhance the functional recovery, they chose R-roscovitine, which can increase pre-synaptic release, based on their single cell RNAseq data and demonstrated that it could further facilitate the behavioral performance in both pre-OPN model and another stroke model. Overall, this manuscript covers a huge amount of high quality data. These results are important because the observed behavioral recovery might have translational potential. I recommend publishing this manuscript.

I have only two minor issues. First, the pre-OPN injury model is novel and could be used by other groups. In light of the complex surgical procedure, it will be useful for the authors to include additional details, such as how to assess complete transection and what are the success rates and so on. Such information will be welcome by the community.

The results in Fig 7 report the effects of the treatment in a stroke model. Comparing to other data, this part is relatively weak. The behavioral assays used here is a sensory test, instead of commonly used motor assays. I would suggest removing this figure and relevant text from this manuscript.

Reviewer #2

(Remarks to the Author)

The authors have used an injury model that lesions the optic tract and tested the use of this model in studying retinofugal regeneration and reconnection. They apply the previously shown approach of knocking down Pten/Socs3 in combination with CNTF expression to show that this manipulation promotes regeneration in the optic tract model. They further show that knocking down Lipin1 further enhances this regeneration and that augmenting presynaptic release or melanopsin overexpression improves partial PLR recovery.

Strengths: The authors have conducted an extensive study using a sophisticated combination of techniques and approaches (anatomical and transsynaptic tracing, PLR, TEM, ephys recordings, and a stroke model) to confirm their

observations and should be applauded for this effort! The main strengths of the manuscript are (i) achieving a complete lesion of the optic tract without damaging the LGN or the OPN, and (ii) achieving PLR recovery and confirming the formation of functional synapses, which is remarkable given the complex wiring of the PLR circuitry with crosstalk from both hemispheres. It is also notable that ipsilateral RGC projections regenerate as well. Together with these two findings, the manuscript indeed presents a powerful model to test the reconnectivity of retinofugal circuits, which is a key current direction for the field of regeneration and visual repair.

Weaknesses: However, since the lesion model is one of the main points of the study, some additional qualitative measures are necessary to deem this lesion as a complete crush injury. And the authors need to carefully reword certain hypotheses and conclusions to avoid misleading the readers.

Major concerns:

1. **Lesion Completeness:** The authors have only used CTb to confirm the lesions – specifically, they report the absence of CTb on the other side of the lesion, and an absence of PLR to conclude the injury as complete. They mostly show coronal orientations in a specific region of the brain. However, this is not sufficient to conclusively confirm that the lesion is complete. The optic tract is wide (~1mm) even in the region where the authors crush and has a certain depth that can lead to many spared axons.

a. It is surprising that the authors have not used two color CTb injections, before and after injury, to confirm completeness. This has been widely used by many papers performing similar experiments. Injecting one colored CTb before injury would be useful to determine that the regions being compared between groups are from the same anatomical regions.

b. While the absence of PLR responses does indicate the axons projecting to the OPN are likely lesioned, there are many other regions adjacent to the OPN that do not depend on PLR and may have spared projections – the NOT, DTN, mdPPN, SC. Alternatively, it is possible that the minimum number of projections needed to produce a PLR response are lesioned. Therefore, the authors could show a few sections of the superior colliculus to confirm the absence of any spared axons/CTb labeling and include either a few images in the horizontal or sagittal planes. The absence of any labeling in the SC would conclusively confirm completeness.

2. **Injury reproducibility:** The authors have used a crush model that has long been debated in the context of the optic nerve crush injury for its completeness and variability in pressure from person to person. There is a need for a more reproducible model, as well as for a lesion that allows the study of target reinnervation.

a. On Page 4, line 1, 5, 6 and so on in the Results section, the authors have used the words “severed, transects, transection” etc. This is completely misleading as the injury described in this manuscript is not severing or transecting (cutting). Please check figure legends and other places too. This is a crush model, and it is very important that the authors check this language to avoid misleading the readers.

b. Authors have NOT provided the stereotaxic coordinates at which they performed the crush injury, thus minimizing its reproducibility. This is particularly surprising since they have included the coordinates used for OPN injections and a few other procedures, except for one of the main findings of the paper. This should be included in the Results and Methods section, as there is no other means to identify the region other than atlas coordinates with respect to Bregma.

c. It is unclear what landmarks the authors use to demarcate the OPN shell versus the core region. These distinctions are typically made using CTb-labeled RGC axonal projections, as there is no specific molecular marker for the OPN or the retinorecipient neurons within the OPN. This is another reason, that a second CTb injected before injury would be useful.

d. There are several images where the lesion site appears to be at variable distances from the dLGN. While the shape of the dLGN changes from anterior to posterior, if the lesion damages the dLGN, this could result in a much more severe injury with a very different regenerative response that could be enhanced or decreased compared to that shown. Such animals should be excluded, and exclusion criteria should be clearly described. Some examples are in Figure 1H between the left and right sides, Figure 1E 6m injury ctrl vs Figure 4E all 3 images shown, Figure 5b top and bottom images, Extended Data figure 1 E vs F, Extended data Figure 4 mouse2 vs mouse 1 or 3.etc..

e. As the authors are focusing on the injury model and comparing it with other studies that induce lesions to the optic nerve and tract, they should cite a more recent study performing a similar injury (Varadarajan et al, Cell Reports 2022 doi: 10.1016/j.celrep.2023.112476 and include an objective comparison in the introduction and discussion.

f. The authors have not addressed the variability in regeneration between animals of the same group, although their images in Extended Data Figure 4 suggest variability. The authors should also include data showing overall regeneration (using normalized Ctb intensity) for each animal and include individual data points for Figure 1G, 1J, Figure 4F, Figure 5C and 5G.

3. **Conclusions from Figure 4A and 4B are misleading and incorrect.** The authors use PRV-GFP to transsynaptically label cells forming connections in the OPN and immunostain with ipRGC markers in the contralateral retina to determine the subtypes of RGCs innervating the OPN. This is theoretically a very clever experiment. However, the conclusions stated from these results are incorrect – “These findings suggest that the ipRGC subtype constitutes the majority of regenerating RGCs forming functional synapses in the OPN” – The PRV virus only labels a small number of cells in the retina (~400 from the authors counts), which may not include the full extent of RGCs innervating the OPN. And since the authors have not immunostained with other RGC markers other than ipRGCs, it is hard to determine using PRV and CTb whether other subtypes of RGCs erroneously innervate the OPN. This conclusion should be removed and corrected to only summarize what the data shows.

Minor Concerns:

1. The authors have a lot of data in the main figures that detracts from the main findings of the study without adding any substantial information. Perhaps, they could consider removing some of these data to focus on the main findings. Some examples: parts of Figure 6, Figure 2 - the TEM data could be moved to extended data, or at the very least include clear demarcation of the PSD and boutons using lines or pseudo-coloring. It is also unclear how PSD length was measured.

2. The authors compare RNA sequencing between the ONC model and their OTI model. While this is very useful, the comparison is shown at 7 days post-injury. This should be clearly stated in the results as a caveat, as this is expected given the location of the crush in the OTI model compared to the ONC model. One would expect cell death to occur later, and as their data in Extended data Figure 2 A-B shows, they do not see any significant cell death until 6 months post-injury.

3. Extended data Figure 4E shows a cross-section of the optic nerve from an Opn4-GFP mouse. It is hard to tell what we are looking at in the first panel and discern whether these are specks or axons. One would expect more GFP+ axons in the optic nerve as this is anterior to the lesion and should still retain intact axons as suggested by the lack of cell death in the retina.

Reviewer #3

(Remarks to the Author)

Zhang, and colleagues investigate brain innervation following axon injury. They developed a new surgical method that involves injuring the optic tract between the lateral geniculate nucleus (LGN) and the olivary pretectal nucleus (OPN). This injury was combined with induced retinal ganglion cell (RGC) axon regeneration using PTEN/SOCS3 deletion and CNTF overexpression. They observed that a few regenerating axons connect to the OPN, enabling some functional recovery. The study also explores enhancing PTEN/SOCS3 deletion and CNTF overexpression induced axon regeneration through RGC activity modulation (as in Lim et al Nat Neuro 2016), lepin1 deletion (Yang et al Neuron 2020), modulation of a calcium-gated channel, and melanopsin overexpression.

This article is well done and could be of interest to the readers. It aligns with previous publications, such as Bei et al., Cell 2016 and Li et al., Neurobiology of Disease 2015: when axons are injured close to their postsynaptic target, some are able to grow back within their targeted brain region and establish functional synapses. In this regard, the novelty and conceptual advance of this work are limited.

Concerns and Suggestions: These are the main points that the authors should address to improve the clarity, depth, and overall impact of their manuscript.

1. Surgical Method Clarification:

- o Please include an image of an injured brain section in Figure 1 (e.g., Fig 1A) to help readers compare the normal and regenerating innervation of the OPN.
- o Please provide the success rate of the surgeries and detailed coordinates for the optic tract injury (OTI) in the methods section. Authors might consider adding a schematic of the surgery in Fig 1 or extended Fig 1 to allow the reproducibility of the surgery.

2. Clarity in Text:

- o Authors should rephrase the sentence on page 4, line 5 ("Our pre-OPN model transects the optic tract right after LGN, severing the RGC axons between LGN and SC/OPN"), to clarify that the LGN is not a relay between the retina and OPN, but rather that RGC axons travel through the optic tract to various brain regions, including the LGN and OPN.
- o Please clarify what is "EW" (Fig3)

3. Experimental Details:

- o Please specify the age of the mice at the start of the experiment (AAV injection) and explain the choice of AAV1-CNTF (p4-line 29) rather than AAV2.

4. Figure Improvements:

- o In extended fig 1F, please provide higher magnification images of the OPN to better appreciate the regenerating axons' innervation patterns. Also, ensure that the OPN is clearly delineated in all figures.
- o Please address the apparent difference in CTB labeling in extended fig 1E and 1F. Is this due to imaging effects?

5. Quantification of Axons:

- o Please quantify the number of axons affected by the OTI to better understand the results.
- o From the extended Fig. 1E-F, it appears that many regenerating axons are following the lesion site as they grow either dorsally or ventrally. Could these phenotypes be quantified to determine the percentage of axons that reach the OPN?

6. RGC Survival and Projections:

- o Please quantify how many RGC axons project beyond the LGN.
- o In Fig3C, it seems that the ventral side (even the naso-ventral) side of the retina mainly project to the OPN. Was RGC survival analyzed in this part of the retina? Compare RGC survival in the specific regions of the retina with the overall retina.
- o Please clarify whether the tract growing beyond the LGN consists of axons or collaterals.
- o In the retina sections, p-Jun expression is still upregulated. Does this occur within RGC projecting beyond the LGN? It would be interesting to label these neurons using specific RGC subpopulation markers, such as co staining of Melanopsin and Brn3b (Chen et al Nature 2011).
- o Are there any survival differences between all ipRGCs and those that primarily project to the OPN?
- o The data concerning the identity of RGCs innervating the OPN are somewhat misleading. Authors report that only 12% of RGCs (relative to intact conditions) innervate the OPN, with 88% of these being ipRGCs and the remaining being alpha RGCs. However, it would be more informative to include absolute numbers as well. Additionally, the graph in Fig. 4A should be revised, as it suggests that the actual numbers (intact vs regeneration) are equivalent.
- o It is already known that only ipRGC and alpha RGC axons regenerate (Tran et al Neuron 2019), so it would be expected to find only these types of axons in the OPN. This point should be addressed in the discussion.

7. Interpretation of RNAseq Data:

The RNAseq data should be discussed in light of the limited number of injured RGC axons, considering the potential impact on the results. If only a subset of RGCs is affected by the OTI, one might not expect to see significant differences compared to the intact condition, whereas stronger differences would be anticipated with the classical ONC, which impacts all RGC.

8. Synapse Formation:

The number of axons entering the OPN should be quantified and compared to the intact condition and provide evidence that GFP-labeled RGCs are exclusively connected to the OPN.

9. Methodological Concerns:

o The choice of using a multi-synaptic virus for tracing is questioned. A monosynaptic rabies virus injected in the OPN might provide more convincing results.

o Controls should be provided to ensure that rabies infection is specific to the OPN and does not involve other brain regions.

10. PLR Experiments:

The delay in response might be due to the limited number of axons activating the OPN, not just the lack of myelination.

Authors might consider comparing the OPN firing rate between intact conditions and 6 months post-injury in the regenerative group.

11. Functional Recovery:

OPN innervation in mice showing PLR recovery 5 months should be quantified after treatment to correlate with functional recovery.

12. Melanopsin Overexpression:

Regarding Melanopsin overexpression: is the endogenous melanopsin expression affected by OTI? In case of overexpression, is melanopsin expressed by other RGC than the ipRGC? Authors should discuss whether melanopsin overexpression affects other RGC and the overall visual system function.

13. Retrograde Labeling:

Please ensure CTB labeling is confined to the OPN without leaking into neighboring areas like the LGN. Authors should include brain sections showing CTB injections to confirm this.

14. Stroke Model and CST Sprouting:

While the stroke model and CST sprouting results are interesting, they don't fit with the main focus of the study. Consider removing these results or expanding their characterization to better integrate them with the main work.

Reviewer #4

(Remarks to the Author)

Axonal regeneration following axotomy is very poor for CNS neurons in general and retinal ganglion cells (RGCs) in particular. However, several groups, most notably that of Zhigang He, have discovered ways to promote regeneration. First, they enhanced mTOR and JAK/STAT signaling by conditional deletion of their endogenous inhibitors, PTEN and SOCS3, from RGCs. Second, they overexpressed CNTF to further boost signaling. Third, they transected the nerve near the target to avoid the need for long-distance regeneration, thus allowing the RGCs axons to grow enough to form synapses with target cells. Fourth, they used a potassium channel blocker to enhance axonal conduction in the face of deficient remyelination. With these methods, they showed synaptogenesis in the superior colliculus and partial recovery of the optomotor reflex (e.g., refs 6, 10, 23, 53, 54).

Zhang et al. have adopted these strategies, but vary the model by crushing the optic tract in a more convenient place, examining synapse formation in a pretectal nucleus (rather than colliculus), and assaying restoration of the pupillary light (rather than optomotor) reflex. They then use this model to (1) document reestablishment of synaptic connectivity by ultrastructural and physiological criteria, with numerous excellent controls (Figures 1-3D); (2) demonstrate behavioral recovery by showing that the pupillary light reflex is partially restored (Figure 3E-H); (3) provide evidence for differential regeneration by distinct RGC types (Figure 4A,B); and (4) describe transcriptional changes that accompany regeneration. They also show that recovery can be enhanced in three different ways: by Lipin 1 knockdown, melanopsin overexpression and activation of presynaptic calcium channels. These additional accomplishments distinguish this paper from those of He et al., so they not only provide a valuable replication but also provide new insights.

There are, however, several problems that need to be addressed:

1) Much is made of the the advantage of the current "between the LGN and the SC" model over Bei's "proximal to the SC" model and the yet to the non-specialist they sound very similar. Some explanation, and perhaps a better diagram is needed to indicate the difference.

Along with the advantages of this model there are a few disadvantages. One is that it requires bilateral rather than unilateral axotomy. Recovery also seems slower – it is evident by 3 months after injury in He's work but only by 6 months here. These are clear from the paper, but should be noted more clearly in the discussion, especially since introduction of the new model is a main point of the paper.

2) "intracranial OTI did not cause noticeable RGC death, in stark contrast with the substantial RGC loss caused by the intraorbital optic nerve crush (ONC)." This dependence of survival with distance has been noted previously, which should be acknowledged.

There is also the complication that not all RGC axons are transected in the model – connections of the LGN and perhaps other retinorecipient areas are maintained. The increased survival is surely in part due to this sparing. In addition, it is possible that intact RGCs can provide support to injured RGCs. In short, these caveats need to be emphasized so the reader

will be able to judge how much confidence to place in the conclusion.

3) The preservation of RGCs also makes the RNAseq data difficult to interpret, since the results are obtained from a combination of axotomized and intact cells. The fraction of “spared” RGCs should be estimated to help the reader understand both the RNAseq and the survival data.

4) Page 13: “Only RGCs from the other eye with axons establishing functional synapses in OPN can be labeled with GFP.” It would take a lot more data to make this statement convincing, since proximity or nonfunctional synapses could also support transfer. This is less convincing than other lines of evidence provided for synaptogenesis – ultrastructure and electrophysiology – and should be toned down or deleted.

5) On specific regeneration of ipRGCs. A caveat here is that M1, M2 and alpha-RGCs are the most successful regenerators (Duan, Neuron, 2015; Tran, Neuron, 2019). Although this preference is decreased in the PTEN/SOCS3/CNTF case it is not abolished (Jacobi, Neuron 2022). So a caveat here is that the selectivity may be partly authentic but also partly a reflection of different regenerative ability.

6) Effects of Lipin knockdown seem noteworthy. However, this reviewer knows nothing about lipin, and no explanation is provided. Background is required – what does it do, what effects on regeneration have been demonstrated previously, why was it chosen and so on. In addition, no data are provided on the efficacy and specificity of lipin knockdown, making it impossible to judge the result.

6) Regarding the lipin result – it is not clear that regeneration is better in the PSL condition (Fig. 5) than when lipin is unaffected (Fig.1) These conditions need to be compared in the same experiment. Similarly, for restoration of function, PSCL needs to be compared with PSC in the same experiment since there may be a rapid recovery in the former between 3 and 6 months. In other words, is it possible that PSC recovers in 3.5 months?

7) Discussion page 12, lines 17-36. Conclusions about mechanism – for example attraction vs repulsion or roles of genes whose expression has not been validated – are stated with excessive confidence. These are interesting speculations, and need to be presented as such.

Version 1:

Reviewer comments:

Reviewer #1

(Remarks to the Author)

In this revised manuscript, the authors have addressed all of the concerns I raised during previous review. This manuscript will be a great addition to the field of axon regeneration.

Reviewer #2

(Remarks to the Author)

The authors have addressed all comments and concerns satisfactorily.

One suggestion remains - to please add sufficient details for the surgical procedure as this is a very good model that will be useful to the field.

The schematic added is good, and it appears they crush in the anteroposterior plane. However, the new details mentioned are unclear as the anteroposterior coordinate is -3.6mm. Presumably the crush injury extends from -0.6 to -3.6mm, i.e. the two arms of the forceps?

Reviewer #3

(Remarks to the Author)

This is a very well done revision. Congratulations!

Reviewer #4

(Remarks to the Author)

The authors have provided an outstandingly comprehensive set of revisions in response to the comments of the reviewers. I congratulate them. Their answers and revisions address my concerns (as reviewer 4) and as far as I can tell, those of reviewers 1-3.

I have three very small suggestions.

1. (related to my point 3) In discussing the role of surviving axons on the RNAseq results, they note that only 10% survive. This is a low but not insignificant number. It should be noted in the text.

2. (related to my point 5) In discussing the selectivity of synapse formation, they note that only 36% of the regenerating axons are ipRGCs or alpha RGS whereas, they see closer to 80% in these categories. There is a clear difference as they note. However, the 36% is already an order of magnitude enrichment over the population as a whole, so this number and the comparison should be noted in the text.

3. The figures for the reviewer are very helpful. I'm having trouble figuring which ones cover material that is also described or illustrated in the article. If there are a few additional panels that could be added to the main or supplementary figures, I'd be very supportive.

REVIEWER COMMENTS

Reviewer #1 (Remarks to the Author):

In this manuscript, Zhang et al report a series of compelling results suggesting that regenerated axons induced by a set of combinatorial treatments (in elevating the intrinsic regenerative ability of adult neurons) could reinnervate physiological targets, make functional synapses and mediate partial behavioral recovery in a nicely designed optic tract injury model. Although previous studies have demonstrated the anatomical regeneration by these treatments, the resultant functional recovery have been difficult to assess. Here the authors first describe an intracranial pre-olivary pretectal nucleus (OPN) injury model. With this, they showed that regenerating axons induced by co-deletion of PTEN/SOCS3 and administration of CNTF are able to reach OPN. By both EM and electrophysiological means, they convincingly show the synapses are formed by these regenerating axons. Importantly, they also showed a partial recovery of pupillary light reflex. Additional over-expression of Lipin1 was able to facilitate axon regeneration and also accelerates behavioral recovery. Importantly, they also demonstrated that these treatments are effective in a chronic model. To further enhance the functional recovery, they chose R-roscovitine, which can increase pre-synaptic release, based on their single cell RNAseq data and demonstrated that it could further facilitate the behavioral performance in both pre-OPN model and another stroke model. Overall, this manuscript covers a huge amount of high quality data. These results are important because the observed behavioral recovery might have translational potential. I recommend publishing this manuscript.

I have only two minor issues. First, the pre-OPN injury model is novel and could be used by other groups. In light of the complex surgical procedure, it will be useful for the authors to include additional details, such as how to assess complete transection and what are the success rates and so on. Such information will be welcome by the community.

Response: Thank you for your insightful suggestion, which could enhance the repeatability of our injury model in the community. We apologize for the oversight in the previous manuscript. We are now incorporating these details into the Results and Methods sections of the manuscript and also attaching them here.

“After the mice were anesthetized, the scalp was cleaned with iodine, and an incision was created to expose the skull. Symmetrical rectangular cranial windows were drilled, and the bone flap was removed to expose cortical tissue. The bilateral optic tracts were completely crushed by forceps (#5 Domont, Fine Science Tools) between the LGN and OPN/SC, the lesion site was situated adjacent to the LGN. Before the surgery, the forceps were carefully checked to ensure that the smooth inner surfaces at the tips could close flat to 5 mm under pressure. During the surgery, the forceps were fixed to the stereotaxic holder with a 3 mm distance between the two tips. The left tip of the forceps was gently placed on the top of the bregma, which was set as the zero point. The forceps were then moved to the coordinates

(anteroposterior -3.6 mm, mediolateral \pm 1.7 mm), and bilateral optic tract crush was performed at a depth of 3.5 mm for 5 seconds. The blood was cleaned, the bones were returned, and the incision was sutured. Meloxicam (1 mg/kg) was subcutaneously injected for analgesia. A heating pad was used to maintain the body temperature of the mice during the surgery until they were fully recovered. The protocols for analgesia and maintaining the mice's body temperature were consistently applied for all subsequent surgeries.

The success of the surgery was confirmed using a two-color CTB injection to assess whether any axons were spared beyond the lesion site. First, CTB-FITC was injected intravitreally in both eyes, followed by bilateral OTI surgery two days later. One day after the OTI, CTB-555 was injected intravitreally in both eyes, and the mice were sacrificed two days later. The two-color CTB labeling was analyzed in the OPN, NOT, mdPPN, and SC, which are located behind the lesion site. The absence of CTB-555 signaling in these nuclei, which had been pre-labeled with CTB-FITC, indicated that the lesion was complete in this surgical model. Mice with LGN damage from the OTI surgery were excluded from further quantification and analysis. The PLR test was also performed on every mouse after injury, and only the mice whose pupils did not respond to light stimulation after pre-OPN OTI indicated the completion of the surgery. The histological and behavioral evaluations indicated that the overall success rates of our OTI model exceed 89%. To examine axon regeneration at various time points after pre-OPN OTI, the control and regeneration mice were sacrificed 2 days after CTB tracing.”

The results in Fig 7 report the effects of the treatment in a stroke model. Comparing to other data, this part is relatively weak. The behavioral assays used here is a sensory test, instead of commonly used motor assays. I would suggest removing this figure and relevant text from this manuscript.

Response: Thank you for your suggestion, the corresponding parts were removed from the revised manuscript.

Reviewer #2 (Remarks to the Author):

The authors have used an injury model that lesions the optic tract and tested the use of this model in studying retinofugal regeneration and reconnection. They apply the previously shown approach of knocking down Pten/Socs3 in combination with CNTF expression to show that this manipulation promotes regeneration in the optic tract model. They further show that knocking down Lipin1 further enhances this regeneration and that augmenting presynaptic release or melanopsin overexpression improves partial PLR recovery.

Strengths: The authors have conducted an extensive study using a sophisticated combination of techniques and approaches (anatomical and transsynaptic tracing, PLR, TEM, ephys recordings, and a stroke model) to confirm their observations and should be applauded for this effort! The main strengths of the manuscript are (i) achieving a complete lesion of the optic tract without damaging the LGN or the OPN, and (ii) achieving PLR recovery and confirming the formation of functional synapses, which is remarkable given the complex wiring of the PLR circuitry with crosstalk from both hemispheres. It is also notable that ipsilateral RGC projections regenerate as well. Together with these two findings, the manuscript indeed presents a powerful model to test the reconnectivity of retinofugal circuits, which is a key current direction for the field of regeneration and visual repair.

Weaknesses: However, since the lesion model is one of the main points of the study, some additional qualitative measures are necessary to deem this lesion as a complete crush injury. And the authors need to carefully reword certain hypotheses and conclusions to avoid misleading the readers.

Major concerns:

1. **Lesion Completeness:** The authors have only used CTB to confirm the lesions – specifically, they report the absence of CTb on the other side of the lesion, and an absence of PLR to conclude the injury as complete. They mostly show coronal orientations in a specific region of the brain. However, this is not sufficient to conclusively confirm that the lesion is complete. The optic tract is wide (~1mm) even in the region where the authors crush and has a certain depth that can lead to many spared axons.

a. It is surprising that the authors have not used two color CTb injections, before and after injury, to confirm completeness. This has been widely used by many papers performing similar experiments. Injecting one colored CTb before injury would be useful to determine that the regions being compared between groups are from the same anatomical regions.

Response: Thank you for your suggestion. To confirm the completeness of the surgery, we conducted the two-color CTB injection as suggested to evaluate whether there were spared axons beyond the lesion site. First, we intravitreously injected CTB-FITC bilaterally, followed by OTI surgery bilaterally two days later. One day after the OTI, we intravitreously injected CTB-555 bilaterally. We compared the CTB labeling in the OPN, NOT, mdPPN, and SC, focusing on areas labeled by CTB-FITC to determine whether they also showed a signal from CTB-555. The results below (Response-Figure 1) showed that there was no CTB-555 signaling

behind the lesion site that had been pre-labeled by CTB-FITC both in coronal sections and sagittal sections. This indicated the completeness of the lesion in our surgery model.

Response-Figure 1: Pre- and post-OTI two-color CTB labeling confirmed the completeness of the injury.

(A-B) Fluorescence images of coronal (A) and sagittal (B) sections showing post-OTI CTB-555 (red) injection did not label RGC axons projecting beyond the lesion site, which was pre-labeled with CTB-FITC (green). All axons projecting beyond the LGN, including those targeting the OPN, NOT, mPPN, and SC, were crushed. Arrowhead indicates the lesion site. Scale bar: (A1-4) 200 μ m, (B) 500 μ m.

b. While the absence of PLR responses does indicate the axons projecting to the OPN are likely lesioned, there are many other regions adjacent to the OPN that do not depend on PLR and may have spared projections – the NOT, DTN, mdPPN, SC. Alternatively, it is possible that the minimum number of projections needed to produce a PLR response are lesioned. Therefore, the authors could show a few sections of the superior colliculus to confirm the absence of any spared axons/CTb labeling and include either a few images in the horizontal or sagittal planes. The absence of any labeling in the SC would conclusively confirm completeness.

Response: Thank you for your suggestion. In the mice we examined shortly after optic tract crush and long-term in the non-regenerated control mice, there were no axons beyond the lesion site in the OPN, SC, or other regions behind the LGN (Supplementary Fig. 1D-1E, and Supplementary Fig. 3A). In our previous manuscript, we only marked the lesion site and LGN. To further confirm the completeness of the surgery, we conducted the two-color CTB injection as suggested to evaluate whether there were spared axons beyond the lesion site. First, we intravitreally injected CTB-FITC bilaterally, followed by OTI surgery bilaterally two days later. One day after the OTI, we intravitreally injected CTB-555 bilaterally. We compared the CTB labeling in the OPN, NOT, mdPPN, and SC, focusing on areas labeled by CTB-FITC to determine whether they also showed a signal from CTB-555. The results (Response-Figure 1) showed that there was no CTB-555 signaling behind the lesion site that had been pre-labeled by CTB-FITC. This indicated the completeness of the lesion in our surgery model.

2. Injury reproducibility: The authors have used a crush model that has long been debated in the context of the optic nerve crush injury for its completeness and variability in pressure from person to person. There is a need for a more reproducible model, as well as for a lesion that allows the study of target reinnervation.

a. On Page 4, line 1, 5, 6 and so on in the Results section, the authors have used the words “severed, transects, transection” etc. This is completely misleading as the injury described in this manuscript is not severing or transecting (cutting). Please check figure legends and other places too. This is a crush model, and it is very important that the authors check this language to avoid misleading the readers.

Response: Thank you for your suggestion, which we fully agree with. We have thoroughly revised the text in the manuscript.

b. Authors have NOT provided the stereotaxic coordinates at which they performed the crush injury, thus minimizing its reproducibility. This is particularly surprising since they have included the coordinates used for OPN injections and a few other procedures, except for one of the main findings of the paper. This should be included in the Results and Methods section, as there is no other means to identify the region other than atlas coordinates with respect to Bregma.

Response: Thank you for your insightful suggestion. We apologize for overlooking this important information in the previous manuscript. We have now included the stereotaxic

coordinates for the crush injury in the Methods section of the manuscript and also attaching them here.

“Before the surgery, the forceps were carefully checked to ensure that the smooth inner surfaces at the tips could close flat to 5 mm under pressure. During the surgery, the forceps were fixed to the stereotaxic holder with a 3 mm distance between the two tips. The left tip of the forceps was gently placed on the top of the bregma, which was set as the zero point. The forceps were then moved to the coordinates (anteroposterior -3.6 mm, mediolateral ± 1.7 mm), and bilateral optic tract crush was performed at a depth of 3.5 mm for 5 seconds.”

c. It is unclear what landmarks the authors use to demarcate the OPN shell versus the core region. These distinctions are typically made using CTb-labeled RGC axonal projections, as there is no specific molecular marker for the OPN or the retinorecipient neurons within the OPN. This is another reason, that a second CTb injected before injury would be useful.

Response: Thank you for your comment. We recognize that a second CTB injection before the injury would be theoretically useful. However, we previously found that the CTB signal does not persist for several months after injection, which prevents us from using a two-color CTB injection in the long-term experiment. For the short-term experiment verifying the completeness of the model injury, we have implemented the suggested two-color CTB strategy and provided the results in the above Response-Figure 1. To distinguish the OPN shell versus the core region, we performed Parvalbumin (PV) staining (Response-Figure 2) according to previous reports^{1,2,3}. This description has been included in the revised manuscript.

1. Osterhout JA, Josten N, Yamada J, Pan F, Wu SW, Nguyen PL, Panagiotakos G, Inoue YU, Egusa SF, Volgyi B, et al. Cadherin-6 mediates axon-target matching in a non-image-forming visual circuit. *Neuron*. 2011;71:632–639.

2. Seabrook TA, Dhande OS, Ishiko N, Wooley VP, Nguyen PL, Huberman AD. Strict Independence of Parallel and Poly-synaptic Axon-Target Matching during Visual Reflex Circuit Assembly. *Cell Rep*. 2017 Dec 12;21(11):3049-3064.

3. Prichard JR, Stoffel RT, Quimby DL, Obermeyer WH, Benca RM, Behan M. Fos immunoreactivity in rat subcortical visual shell in response to illuminance changes. *Neuroscience*. 2002;114(3):781-93.

Response-Figure 2: Parvalbumin staining to delineate the OPN region.

(A) Representative images of Parvalbumin (PV) staining (red) in brain sections from sham and 6 m Regen. group. The shown brain regions were contralateral to the CTB-FITC (green) injected eye. Scale bar: 100 μ m.

d. There are several images where the lesion site appears to be at variable distances from the dLGN. While the shape of the dLGN changes from anterior to posterior, if the lesion damages the dLGN, this could result in a much more severe injury with a very different regenerative response that could be enhanced or decreased compared to that shown. Such animals should be excluded, and exclusion criteria should be clearly described. Some examples are in Figure 1H between the left and right sides, Figure 1E 6m injury ctrl vs Figure 4E all 3 images shown, Figure 5b top and bottom images, Supplementary figure 1 E vs F, Supplementary Figure 4 mouse2 vs mouse 1 or 3.etc..

Response: Thank you for your suggestion. We have recognized this important issue, even though we observed only a limited number of mice with LGN damage among the examined mice. Therefore, we included only animals with undamaged LGN in the results presented in the previous manuscript. However, as you suggested, we should clarify these exclusion criteria in the manuscript, and we have updated the Methods section accordingly (“Mice with LGN damage from the OTI surgery were excluded from further quantification and analysis”). We also included one animal with dLGN damage here (Response-Figure 3) for your comparison with those undamaged ones. Regarding the images mentioned in your question, we hope you can understand that there may be variations between different animals concerning the distances between lesion sites and dLGN, which could arise from differences in brain sizes.

Response-Figure 3: Example of OTI exclusion due to damaging LGN.

Representative images showing the comparison between LGN-damaged OTI and LGN-undamaged OTI. CTB-FITC (green) was injected intravitreally to label the RGC axons. GFAP was used for indicating the lesion site (red). LGN was outlined with dashed lines and arrowheads indicated the lesion sites. Scale bar: 200 μ m.

e. As the authors are focusing on the injury model and comparing it with other studies that induce lesions to the optic nerve and tract, they should cite a more recent study performing a similar injury (Varadarajan et al, Cell Reports 2022 doi: 10.1016/j.celrep.2023.112476 and include an objective comparison in the introduction and discussion.

Response: Thank you for your suggestion. We have cited this paper in the manuscript and included an objective comparison in the introduction section.

f. The authors have not addressed the variability in regeneration between animals of the same group, although their images in Supplementary Figure 4 suggest variability. The authors should also include data showing overall regeneration (using normalized Ctb intensity) for each animal and include individual data points for Figure 1G, 1J, Figure 4F, Figure 5C and 5G.

Response: Thank you for your suggestion. We observed variability in the number of regenerated axons in the OPN region due to multiple factors, such as AAV expression and variations in injury. However, we included a sufficient number of animals in each treatment group, and all animals in the regeneration group showed axon regeneration and reinnervation of the OPN region at 6 months post-injury in the PSC group (Pten/Socs3 knockout and CNTF expression) and at 3 months post-injury in the PSCL group (combination of Lipin1 knockdown

with Pten/Socs3 knockout and CNTF expression). We have included individual data points to illustrate axon regeneration in the statistical graphics.

3. Conclusions from Figure 4A and 4B are misleading and incorrect. The authors use PRV-GFP to transsynaptically label cells forming connections in the OPN and immunostain with ipRGC markers in the contralateral retina to determine the subtypes of RGCs innervating the OPN. This is theoretically a very clever experiment. However, the conclusions stated from these results are incorrect – “These findings suggest that the ipRGC subtype constitutes the majority of regenerating RGCs forming functional synapses in the OPN” – The PRV virus only labels a small number of cells in the retina (~400 from the authors counts), which may not include the full extent of RGCs innervating the OPN. And since the authors have not immunostained with other RGC markers other than ipRGCs, it is hard to determine using PRV and CTb whether other subtypes of RGCs erroneously innervate the OPN. This conclusion should be removed and corrected to only summarize what the data shows.

Response: Thank you for your comment. The PRV transsynaptic labeling experiment only identified the RGC types that formed synapses with the OPN neurons that project to the EW nucleus, not those that regenerated without forming synapses. Therefore, our conclusion primarily focuses on the RGC types that reform synapses with OPN neurons. We have rephrased our conclusion in the manuscript.

“These findings suggest that the ipRGC subtype constitutes the majority of RGCs that form synapses with the OPN neurons related to the PLR response in the Regen. mice.”

Minor Concerns:

1. The authors have a lot of data in the main figures that detracts from the main findings of the study without adding any substantial information. Perhaps, they could consider removing some of these data to focus on the main findings. Some examples: parts of Figure 6, Figure 2 - the TEM data could be moved to Supplementary, or at the very least include clear demarcation of the PSD and boutons using lines or pseudo-coloring. It is also unclear how PSD length was measured.

Response: Thank you for your suggestions. We decided to move the original Figures 6A-6D to supplementary Fig. 13. We would like to keep the original Figures 6E-6I because we think they are relevant important findings. As TEM data (Figure 2) represents the gold standard for assessing synapses morphologically, we decided to keep this part. We have delineated the TEM data with symbols and pseudo-coloring as you suggested.

We have included the method for measuring PSD length in the manuscript and are also attaching them here.

“After capturing the EM images, the length of the PSD was measured manually using ImageJ. For quantification, only synapses with clearly defined structures were included. The PSD length was measured by manually drawing a line along the PSD in the postsynaptic membrane, utilizing the calibrated scale of the EM images.”

2. The authors compare RNA sequencing between the ONC model and their OTI model. While this is very useful, the comparison is shown at 7 days post-injury. This should be clearly stated in the results as a caveat, as this is expected given the location of the crush in the OTI model compared to the ONC model. One would expect cell death to occur later, and as their data in Supplementary Figure 2 A-B shows, they do not see any significant cell death until 6 months post-injury.

Response: Thank you for your comment. Actually, we did not observe any RGC death even 6 months post-OTI (Supplementary Fig. 5A-5B). We primarily focused on the early stages post-injury because the changes in gene expression after optic nerve injury are more pronounced during this time¹. Through this kind of comparison, we correlate the RGC cell death differences between the two models with different levels of cell death signal activation downstream of the DLK pathway. Our hypothesis is that the weaker activation of this pathway is the main reason for nearly no cell death observed in the OTI model. Though, indeed, we agree that increasing longer time points for comparison will provide more information.

1. Jacobi A, Tran NM, Yan W, Benhar I, Tian F, Schaffer R, He Z, Sanes JR. Overlapping transcriptional programs promote survival and axonal regeneration of injured retinal ganglion cells. *Neuron*. 2022 Aug 17;110(16):2625-2645.e7.

3. Supplementary Figure 4E shows a cross-section of the optic nerve from an *Opn4-GFP* mouse. It is hard to tell what we are looking at in the first panel and discern whether these are specks or axons. One would expect more GFP⁺ axons in the optic nerve as this is anterior to the lesion and should still retain intact axons as suggested by the lack of cell death in the retina.

Response: Thank you for your comments. The *Opn4-GFP* mouse line we used labels only M1-M3 ipRGCs, not all ipRGCs¹. In the following attached Response-Figure 4 (A, B), we show that GFP⁺ RGCs only constitute around 4% of all the RGCs in the *Opn4-GFP* mouse. We also co-stained a longitudinal sections and cross-section of the optic nerve with a Tuj1 antibody to confirm that the GFP-positive signals are indeed axons (C, D). To enhance the visibility of the GFP signals, we have replaced the images with high-power views of the GFP, Tuj1, and MBP triple staining in the revised manuscript (Supplementary Fig. 11E).

1. Li S, Yang C, Zhang L, Gao X, Wang X, Liu W, Wang Y, Jiang S, Wong YH, Zhang Y, Liu K. Promoting axon regeneration in the adult CNS by modulation of the melanopsin/GPCR signaling. *Proc Natl Acad Sci U S A*. 2016 Feb 16;113(7):1937-42.

Response-Figure 4: Characteristics of the Opn4-GFP mice.

(A) Representative images of the flatmount retina with Tuj1 (blue) and GFP (green) signals from Opn4-GFP mice. Scale bar: 50 μ m.

(B) Quantification of the percentage of GFP+ RGCs in Tuj1+ RGCs for (A). Data presents mean \pm SEM. n=5 mice.

(C-D) Longitude and cross-sections of the optic nerve from Opn4-GFP mice showing co-staining of GFP (green) and Tuj1 (blue), indicating GFP-positive signals are indeed axons. Scale bar: C(i) 10 μ m, C(ii) 2 μ m, (D) 2 μ m. C(ii) is the enlargement of the rectangle dotted-line labeled area in C(i). White arrows label the GFP+ Tuj1+ axons.

Reviewer #3 (Remarks to the Author):

Zhang, and colleagues investigate brain innervation following axon injury. They developed a new surgical method that involves injuring the optic tract between the lateral geniculate nucleus (LGN) and the olivary pretectal nucleus (OPN). This injury was combined with induced retinal ganglion cell (RGC) axon regeneration using PTEN/SOCS3 deletion and CNTF overexpression. They observed that a few regenerating axons connect to the OPN, enabling some functional recovery. The study also explores enhancing PTEN/SOCS3 deletion and CNTF overexpression induced axon regeneration through RGC activity modulation (as in Lim et al Nat Neuro 2016), lepin1 deletion (Yang et al Neuron 2020), modulation of a calcium-gated channel, and melanopsin overexpression.

This article is well done and could be of interest to the readers. It aligns with previous publications, such as Bei et al., Cell 2016 and Li et al., Neurobiology of Disease 2015: when axons are injured close to their postsynaptic target, some are able to grow back within their targeted brain region and establish functional synapses. In this regard, the novelty and conceptual advance of this work are limited.

Concerns and Suggestions: These are the main points that the authors should address to improve the clarity, depth, and overall impact of their manuscript.

1. Surgical Method Clarification:

o Please include an image of an injured brain section in Figure 1 (e.g., Fig 1A) to help readers compare the normal and regenerating innervation of the OPN.

Response: Thank you for your suggestion. We have included an uninjured brain section in Supplementary Fig. 1A, while the brain sections for the injured short-term, long-term non-regenerated control, and long-term regenerated conditions are presented in Supplementary Fig. 1D, Supplementary Fig. 3, Supplementary Fig. 11A, Fig. 1E, and Fig. 1H.

o Please provide the success rate of the surgeries and detailed coordinates for the optic tract injury (OTI) in the methods section. Authors might consider adding a schematic of the surgery in Fig 1 or extended Fig 1 to allow the reproducibility of the surgery.

Response: Thank you for your suggestions. We have included the surgery success rate and detailed coordinates for the OTI in the methods section. Additionally, a scheme of the surgery was added as suggested (see Video S1 and the following Response-Figure 5).

“Before the surgery, the forceps were carefully checked to ensure that the smooth inner surfaces at the tips could close flat to 5 mm under pressure. During the surgery, the forceps were fixed to the stereotaxic holder with a 3 mm distance between the two tips. The left tip of the forceps was gently placed on the top of the bregma, which was set as the zero point. The forceps were then moved to the coordinates (anteroposterior -3.6 mm, mediolateral ± 1.7 mm), and bilateral optic tract crush was performed at a depth of 3.5 mm for 5 seconds.”

“The histological and behavioral evaluations indicated that the overall success rates of our OTI model exceed 89%.”

Response-Figure 5: Scheme of the OTI model.

(A) Showing are some key steps (3-6) and different views (1-2, 7-9) of the OTI from one side.

2. Clarity in Text:

o Authors should rephrase the sentence on page 4, line 5 (“Our pre-OPN model transects the optic tract right after LGN, severing the RGC axons between LGN and SC/OPN”), to clarify that the LGN is not a relay between the retina and OPN, but rather that RGC axons travel through the optic tract to various brain regions, including the LGN and OPN.

Response: Thank you for your suggestion. We have rephrased the sentence.

“In intact mice, the axons of RGCs extend from the retina into the brain, sequentially forming the optic nerve, the optic chiasm, then through the optic tract to reach different brain nuclei, including LGN, OPN, SC, and others. Our pre-OPN model crushes the optic tract right after LGN, injuring the RGC axons between LGN and OPN/SC.”

o Please clarify what is “EW” (Fig3)

Response: EW is an abbreviation for the Edinger–Westphal nucleus. EW receives afferents from OPN and is involved in the pupillary light reflex circuit. We have clarified this in the manuscript.

“In the multisynaptic tracing experiment, after the injection of PRV-GFP into the anterior chamber of one eye, it passes through the ciliary body to retrogradely infect ciliary ganglion cells, then reaching the Edinger-Westphal nucleus (EW) and OPN. The EW receives afferents from the OPN and is involved in the PLR circuit. RGCs from the other eye, whose axons establish synapses with OPN neurons that project to the EW, can be labeled with GFP.”

3. Experimental Details:

o Please specify the age of the mice at the start of the experiment (AAV injection) and explain the choice of AAV1-CNTF (p4-line 29) rather than AAV2.

Response: Thank you for your suggestions. We have included information about the age of the mice in the corresponding Methods section and figures. The AAV-CNTF we used is AAV2, this is a typo, and we have corrected it in the manuscript.

4. Figure Improvements:

o In extended fig 1F, please provide higher magnification images of the OPN to better appreciate the regenerating axons' innervation patterns. Also, ensure that the OPN is clearly delineated in all figures.

Response: Thank you for your suggestions. We have included higher magnification images of the OPN, which are now presented in Supplementary Fig. 3C of the revised manuscript. Additionally, we have delineated the OPN in all figures in the revised manuscript.

o Please address the apparent difference in CTB labeling in extended fig 1E and 1F. Is this due to imaging effects?

Response: Thank you for your comment. This is due to the differences in image contrast and brightness. We have adjusted these parameters to ensure that the two panels appear similar.

5. Quantification of Axons:

o Please quantify the number of axons affected by the OTI to better understand the results.

Response: Thank you for your suggestion. Since the OTI crushes all axons beyond the LGN (see Supplementary Fig. 1D-1E), directly quantifying the number of axons affected by the OTI is challenging due to the large number of axons involved. Therefore, we used an indirect approach to label the RGCs that project their axons beyond the LGN, which would be injured by the OTI. We performed retrograde labeling by injecting CTB555 into the SC and OPN of intact C57BL/6J mice (Response-Figures 6A-C). This method allows us to quantify the percentage of RGCs projecting to the SC and OPN, representing the minimum proportion of RGCs affected by the OTI. The results indicate that more than 90% of RGCs are CTB555 positive and were injured by the OTI (Response-Figures 6D-E).

Response-Figure 6: Retrograde labeling of RGCs that project beyond LGN.

(A-C) Representative images showing the injection sites of the retrograde tracer CTB555 (red) (A, OPN; B, enlarged OPN and LGN region; C, SC). Scale bar: (A) 200 μ m, (B) 100 μ m, (C) 200 μ m.

(D) Representative images of the flat-mounted retina showing the retrogradely labeled RGCs by CTB555 (red) were colocalized with RGC marker Tuj1 (green). Scale bar: 50 μ m.

(E) Quantification of the percentage of the retrogradely labeled RGCs that project beyond LGN for (D). Data presents mean \pm SEM. n=3 mice.

o From the extended Fig. 1E-F, it appears that many regenerating axons are following the lesion site as they grow either dorsally or ventrally. Could these phenotypes be quantified to determine the percentage of axons that reach the OPN?

Response: Thank you for your suggestion. We quantified the percentage of axons that grow dorsally toward the cortex, ventrally toward the thalamus, or along their original trajectory toward the OPN by measuring the CTB intensity within 200 μ m of the lesion site (Response-Figure 7), as nearly all regenerated axons growing dorsally toward the cortex or ventrally could only be detected within this distance.

A

Response-Figure 7: Quantification of regenerating axons along different directions.

(A) Quantification of regenerating axons by normalized CTB intensity along dorsal, ventral, or original trajectory toward the OPN for animals in Figures 1E-1G. Data presents mean \pm SEM. n=8 mice. (***) $p \leq 0.001$, ANOVA followed by Bonferroni test)

6. RGC Survival and Projections:

o Please quantify how many RGC axons project beyond the LGN.

Response: Thank you for your suggestion. Similar to Question 5, directly quantifying the number of RGC axons projecting beyond the LGN is challenging due to the large number of axons involved. Therefore, we used an indirect approach to retrograde label the RGCs that project their axons beyond the LGN. We performed retrograde labeling by injecting CTB555 into the SC and OPN of intact C57BL/6J mice. This method allowed us to quantify the percentage of RGCs projecting to the SC and OPN, representing the minimum proportion of RGCs that project axons beyond the LGN. The results (Response-Figure 6) indicated that more than 90% of RGCs project their axons beyond the LGN.

o In Fig3C, it seems that the ventral side (even the naso-ventral) side of the retina mainly project to the OPN. Was RGC survival analyzed in this part of the retina? Compare RGC survival in the specific regions of the retina with the overall retina.

Response: RGC survival was assessed throughout the entire retina. Although the RGCs that connect to the OPN are mainly found in the naso-ventral region, the OTI also affects RGCs that send axons to the OPN, SC and other areas beyond the LGN. As shown in the figure in response to Question 5, over 90% of RGCs are injured by the OTI. Furthermore, we did not detect differences in RGC survival across the different regions of the retina.

o Please clarify whether the tract growing beyond the LGN consists of axons or collaterals.

Response: Thank you for your suggestion. Our OTI is a complete injury model that damages all axons beyond the LGN (see Supplementary Fig. 1D-1E). Axons identified beyond the lesion site in the Regen. group are regenerated axons. Thus, there are no axon collaterals from the spared axons. We took high-magnification images post the lesion site to examine the side branches of the regenerated axons and found that collaterals are present in the regenerated axons, as shown in the following figure (Response-Figure 8).

Response-Figure 8: Regenerating axons beyond LGN consist of collaterals.

(A-B) Two examples showing regenerating axons beyond LGN consist of collaterals. The right panels label the outline of regenerated axons with dashed lines. Scale bar: (A) 5 μ m, (B) 10 μ m.

o In the retina sections, p-Jun expression is still upregulated. Does this occur within RGC projecting beyond the LGN? It would be interesting to label these neurons using specific RGC subpopulation markers, such as co staining of Melanopsin and Brn3b (Chen et al Nature 2011).

Response: Thank you for your suggestion. We injected CTB555 into the SC and OPN of intact C57BL/6J mice to label RGCs that project axons beyond the LGN. Three days later, we performed the OTI. We checked the p-c-Jun levels in these CTB-labeled RGCs five days after the OTI (Response-Figure 9). The quantification showed that nearly all (98%) p-c-Jun+ RGCs are CTB555+ RGCs, indicating that p-c-Jun expression is upregulated only in RGCs that project beyond the LGN and were injured by the OTI (Response-Figure 9A). Besides, p-c-Jun was barely detected in the Mela.+ or Brn3b+ RGCs (Response-Figures 9B-C). Since we want to continue an independent study on the subtypes of RGCs that show different responses to injury, we have not included the Response-Figures 9B-C in the revised manuscript. If the reviewer finds this data necessary, we can add it.

Response-Figure 9: p-c-Jun expression was upregulated only in RGCs that were injured by the OTI.

(A) Representative images illustrate the colocalization of p-c-Jun (blue) and OTI-affected RGCs labeled by CTB555 (red) in mice at 5 days post-OTI. Tuj1 staining (green) highlights all RGCs. White arrows indicate OTI-activated p-c-Jun+ RGCs co-localize with CTB555+ RGCs. The right panel shows the quantification of the percentage of CTB555+ RGCs among p-c-Jun+ RGCs. Data presents mean \pm SEM. n=3 mice. Scale bar: 50 μ m.

(B-C) Representative images show the colocalization of p-c-Jun (blue), CTB555 (red), and Melanopsin (B)/Brn3b (C) (green) in sectioned retinas from 5-day OTI mice. The yellow arrows indicate p-c-Jun+ RGCs that colocalize with CTB555 but not with Melanopsin (B) or Brn3b (C). Scale bar: 50 μ m.

o Are there any survival differences between all ipRGCs and those that primarily project to the OPN?

Response: There is nearly no cell death in the entire RGC population after OTI (Response-Figure 10A). We also checked that Tbr2+ RGCs (Response-Figure 10B), melanopsin+ ipRGCs (Response-Figure 10C), and SMI32+ alpha-RGCs (Response-Figure 10D) showed no survival differences between intact and post-OTI conditions. Based on these data, we speculate that there are no survival differences between all ipRGCs and those that primarily project to the OPN.

Response-Figure 10: Cell survival in OTI for all RGCs and subtype-specific RGCs.

(A-D) Representative images showing the flat-mounted retina staining with pan-RGC maker Tuj1 (green A), and subtype-specific markers Tbr2 (purple B), Melanopsin (red C), SMI32 (cyan D) in different times post-OTI and corresponding quantification of the cell density. Data presents mean \pm SEM. n=3-4 mice. Scale bar: 50 μ m.

o The data concerning the identity of RGCs innervating the OPN are somewhat misleading. Authors report that only 12% of RGCs (relative to intact conditions) innervate the OPN, with 88% of these being ipRGCs and the remaining being alpha RGCs. However, it would be more informative to include absolute numbers as well. Additionally, the graph in Fig. 4A should be revised, as it suggests that the actual numbers (intact vs regeneration) are equivalent.

Response: Thank you for your suggestion. The absolute numbers of different RGC types are included in Supplementary Fig. 8F, also attached here (Response-Figure 11). We used percentages in Fig 4A to illustrate the composition of different RGC types, as absolute numbers may not clearly convey the cell type distribution.

Response-Figure 11: Quantification of the number of different types of RGC in PRV-GFP traced cells.

Quantification of the number of PRV-GFP-traced melanopsin+ ipRGCs, SMI32+ α RGCs, and other RGC types. Data presents mean \pm SEM. n=3 mice. (ns, not significant $p > 0.05$, * $p \leq 0.05$, ANOVA followed by Bonferroni test)

o It is already known that only ipRGC and alpha RGC axons regenerate (Tran et al Neuron 2019), so it would be expected to find only these types of axons in the OPN. This point should be addressed in the discussion.

Response: Thank you for your suggestion. The study mentioned here (Tran et al Neuron 2019) only applies to optic nerve injury in wild-type mice without manipulation. Another paper “Overlapping transcriptional programs promote survival and axonal regeneration of injured retinal ganglion cells” from the same group showed that Pten/Socs3 deletion combined with CNTF could overcome the type-specific barriers for RGC axon regeneration. Although we utilized a different injury model, we would speculate that other RGC types, in addition to ipRGCs and alpha RGCs, may also regenerate their axons because we adopted a similar genetic manipulation.

However, instead of focusing on which RGC subtypes regenerate more effectively, as has already been addressed by other groups using the optic nerve crush model, our study utilizes the OTI model to identify the RGC subtypes that reestablish synapses with target neurons, a process more directly relevant to functional recovery. Thus, we conducted trans-synaptic tracing experiments using PRV-GFP. This method does not identify all RGC types that regenerate their axons; it only identifies those cell types that form synapses with the OPN neurons projecting to the EW nucleus. Therefore, our results do not suggest that only ipRGC and alpha RGC axons regenerated to the OPN.

7. Interpretation of RNAseq Data:

The RNAseq data should be discussed in light of the limited number of injured RGC axons, considering the potential impact on the results. If only a subset of RGCs is affected by the OTI, one might not expect to see significant differences compared to the intact condition, whereas stronger differences would be anticipated with the classical ONC, which impacts all RGC.

Response: Thank you for your comment. The OTI crushes all axons beyond the LGN (Supplementary Fig. 1D-1E), affecting over 90% of RGCs according to the CTB555 retrograde labeling result in response to Question 5. Therefore, the differences observed should not be due to a limited number of RGCs being injured after OTI, but rather due to the distance of the injury, intact axon collaterals before the lesion site, or other reasons.

“The striking difference in RGC survival is only partially attributable to the number of RGCs that are injured, as the OTI model affects over 90% of RGCs, according to the CTB-555 retrograde labeling results (Supplementary Fig. 2D-2E). Instead, the distance from the lesion site and the presence of intact branches before the lesion site appear to be more significant factors, as previously reported^{18,38-40}”

8. Synapse Formation:

The number of axons entering the OPN should be quantified and compared to the intact condition and provide evidence that GFP-labeled RGCs are exclusively connected to the OPN.

Response: Thank you for your suggestions. Since it is difficult to quantify the exact number of axons entering the OPN due to their high density, especially in the intact condition, we instead quantified the number of CTB-labeled axon terminal boutons in the OPN for both the regeneration group and the Sham condition. As shown in the figure below (Response-Figure 12), Sham mice have around 90 boutons/100 μm^2 , while regeneration mice have around 10 boutons/100 μm^2 .

Response-Figure 12: Axon terminal boutons within the OPN region.

(A) Representative images showing the CTB-FITC-labeled RGC axon terminal boutons (green) in OPN regions from sham and 6m Regen. groups. Scale bar: 10 μ m.

(B) Quantification of density of CTB-labeled boutons in OPN from sham and 6 m Regen. groups. Data presents mean \pm SEM. n=3 mice. (** $p \leq 0.01$, student's t-test)

To ensure that GFP-labeled RGCs are exclusively connected to the OPN, we checked the GFP-labeled neurons in the brains of 6 m Injury Ctrl. mice 5 days following PRV injection (Response-Figure 13). We found that GFP was present in the EW and OPN, which are located downstream of the lesion site in the PLR circuit, but not in the LGN and suprachiasmatic nucleus (SCN) regions, which are upstream of the lesion site. This indicates that the GFP expressed in RGCs in the regeneration group results from transsynaptic transport between OPN neurons and regenerated RGC axons within the OPN.

Response-Figure 13: PRV-labeled brain regions in 6 m Injury Ctrl. Mice.

(A-C) Representative images showing PRV-GFP (green) retrogradely traced brain nuclei in the major RGC projection regions of the 6 m Injury Ctrl. mice, outlined with dashed lines. These regions include the SC (A), EW (A), OPN (B), LGN (B), and SCN (C). White arrowheads indicate the lesion site in (B). Scale bars: (A) 500 μ m, (B) 200 μ m, (C) 200 μ m.

(D-E) Representative zoomed-in images showing PRV-GFP-labeled cells in the EW (i, D) and OPN (ii, E). Scale bars: (D) 100 μ m, (E) 100 μ m.

9. Methodological Concerns:

- o The choice of using a multi-synaptic virus for tracing is questioned. A monosynaptic rabies virus injected in the OPN might provide more convincing results.
- o Controls should be provided to ensure that rabies infection is specific to the OPN and does not involve other brain regions.

Response: Thank you for your suggestions. We agree that using the monosynaptic rabies virus could further support the findings from the previous PRV-GFP results. Therefore, we conducted monosynaptic rabies virus injections as follows: In sham mice, non-regenerated control mice, and regenerated mice, the helper viruses (AAV9-Ef1a-DIO-His-EGFP-2a-TVA-WPRE-pA, AAV9-Ef1a-DIO-oRVG-WPRE-Pa), and AAV1-Syn-cre were injected into the OPN. Three weeks later, RV-ENVA- Δ G-dsRed was injected into the same OPN region, and RGCs were examined another week later. The results (Response-Figure 14) showed that there were no dsRed-labeled RGCs found in the retinas of 6 m Injury Ctrl. mice. In Sham mice, an average of 1304 dsRed-labeled RGCs per retina were identified, whereas the average number of labeled RGCs in the 6 m Regen. mice were 89.

Please note that the multisynaptic retrograde tracing using PRV-GFP can only label the RGCs whose axons re-establish synapses with OPN neurons projecting to the EW, as the virus was delivered to the anterior chamber of the eye to infect the iris, which is downstream of OPN in the PLR circuit. In contrast, monosynaptic rabies virus tracing will label all RGCs that reestablish synapses with OPN neurons, regardless of their relation to the PLR response.

Response-Figure 14: Monosynaptic rabies virus tracing of functional synapses forming RGCs.

(A) A diagram illustrating the steps and labelling outcomes of retrograde monosynaptic tracing using the rabies virus. The injection of AAV-Cre and AAV-Helper (AAV-DIO-EGFP-TVA, AAV-DIO-RVG) results in the labeling of the 0⁰ OPN neurons with GFP. Following this, the injection of RV-dsRed labels both the 0⁰ OPN neurons and the 1⁰ RGCs, which forms a monosynaptic connection with the 0⁰ OPN neurons, with dsRed, while the 2⁰ neuron remains unlabeled.

(B) A schematic representation illustrates rabies virus retrograde tracing from the OPN to the cell bodies of RGCs in the eye injected with AAV-CNTF/Cre. The rabies virus identifies RGCs whose regenerated axons successfully extended to the OPN and reformed synaptic connections.

(C-D) Fluorescence images depict the injection site in Sham and 6 m Injury Ctrl. mice. The helper viruses and AAV1-Syn-Cre were injected into the OPN of the mice (green). Three weeks after the helper virus injection, RV-EnvA-ΔG-dsRed (red) was injected into the OPN. The arrowhead indicates the lesion site in (D). Scale bar: 200 μm.

(E-F) Magnified images of the areas indicated by rectangles in (C) and (D), showing the injection site in the OPN region. Scale bar: 100 μm.

(G) Fluorescence images display RV-dsRed-traced cells (red) co-localized with the RGC marker Tuj1 (green) in Sham and 6 m Regen. mice. Scale bar: 20 μm.

(H) Flat-mount retina images show RV-traced RGCs (black dots) across different groups. Scale bar: 1000 μm. The lower panel presents the average number of traced RGCs in each group.

10. PLR Experiments:

The delay in response might be due to the limited number of axons activating the OPN, not just the lack of myelination. Authors might consider comparing the OPN firing rate between intact conditions and 6 months post-injury in the regenerative group.

Response: Thank you for your suggestion. We agree that the delay in response might also be due to the limited number of axons activating the OPN. We have recorded the firing rate in Sham group and 6 months post-injury in the regenerative group, and the result (Response-Figure 15) showed that light-induced OPN neuronal firing rate increased in the 6 m Regen. group was not as strong as in the Sham group. We integrated this data in the revised manuscript.

“In Sham mice, light stimulation significantly increased OPN neuronal firing rate (Fig. 3D). In contrast, light stimulation failed to increase neuronal firing rate in the OPN of the 6 m Injury Ctrl. mice. However, light significantly boosted the neuronal responses in 6 m Regen. mice, although not as strongly as in the Sham condition (Fig. 3D and Supplementary Fig. 10D), suggesting the existence of functional synaptic connections between the regenerated RGC axons and OPN.”

Response-Figure 15: Quantification of the number of spikes per second (firing rate) in the OPN before and after 1 minute blue light (0.5 mW/cm²) stimulation in the sham group, 6 m Injury Ctrl. group, and 6 m Regen. group. The microelectrode array was inserted into the contralateral OPN of the AAV-injected eye. Data presents mean \pm SEM. n=6 in each group. (ns, not significant $p > 0.05$, * $p \leq 0.05$, ** $p \leq 0.01$, multiple paired t-tests)

11. Functional Recovery:

OPN innervation in mice showing PLR recovery 5 months should be quantified after treatment to correlate with functional recovery.

Response: Thank you for your suggestion. We observed that some mice with PSC treatment (Pten/Socs3 knockout and CNTF expression) recovered their PLR within 5 months, but most did not (Response-Figure 16). Therefore, we extended the observation period by one month and found that PLR recovery was consistently observed in all animals (Fig. 3F). Thus, we decided to set 6 months post-injury as a time point to assess axon regeneration. We did not have a batch of animals terminated at 5 months post-injury to evaluate their regeneration, and preparing a new batch of experiments requires a significant amount of time, approximately 8 months. On the other hand, we demonstrated the correlation between OPN innervation of regenerated axons and functional recovery in two aspects. Firstly, at 3 months post-OTI in the PSC group, we did not observe regenerated axons growing into the OPN region, which is consistent with the lack of PLR recovery detected at this time point. Secondly, at 6 months post-OTI, all mice that showed PLR recovery had their axons reinnervate the OPN, as shown in Supplementary Fig. 11A and 11B, in which there is a strong correlation between OPN reinnervation and functional recovery. Therefore, we hope the reviewers and editors could understand that we did not re-prepare the experiments.

Response-Figure 16: Quantification of the PLR following one minute blue light (0.5 mW/cm²) stimulation in the Regen. group (Pten/Socs3 knockout combined with CNTF expression) at four and five months after pre-OPN OTI for animals in Figure 3F. No PLR response was observed in 4 m Regen. mice. However, two out of eight mice exhibited PLR response to light stimulation starting at 5 months post-OTI.

12. Melanopsin Overexpression:

Regarding Melanopsin overexpression: is the endogenous melanopsin expression affected by OTI? In case of overexpression, is melanopsin expressed by other RGC than the ipRGC? Authors should discuss whether melanopsin overexpression affects other RGC and the overall visual system function.

Response: Thank you for your suggestion. We performed melanopsin staining in both optic tract crush mice and intact mice and found no differences in the percentage of melanopsin+ RGCs between the groups (Response-Figure 10C).

As shown in the following figure, after AAV-Melanopsin injection, around 80% of RGCs expressed melanopsin, compared to only 5% in non-AAV-injected wild-type mice (Response-Figures 17A-B). This indicates that other types of RGCs than ipRGCs also expressed melanopsin. We conducted co-staining of melanopsin with *Tbr1*, which is the T-RGC marker (Tran, Nicholas M., et al. "Single-cell profiles of retinal ganglion cells differing in resilience to injury reveal neuroprotective genes." *Neuron* 104.6 (2019): 1039-1055.) and found that T-RGCs also expressed melanopsin in the melanopsin overexpression group (Response-Figure 17C).

Since this study focuses on the non-image-forming visual function, specifically the PLR, we conducted a PLR test in intact mice with and without melanopsin overexpression. We initially tested the PLR under light stimulation at the same intensity (0.5 mW/cm²) used in other experiments in this paper. The pupil constriction dynamics within 1 minute of light stimulation were enhanced after melanopsin overexpression (Response-Figures 17D-E). Since melanopsin overexpression increased the sensitivity of the RGCs^{1,2}, the difference in pupil constriction under low light intensity conditions may also be pronounced. Indeed, melanopsin overexpression more obviously enhanced the PLR at a light intensity of 0.01 mW/cm² (Response-Figures 17F-G).

1. Li, S. et al. Promoting axon regeneration in the adult CNS by modulation of the melanopsin/GPCR signaling. *Proc Natl Acad Sci U S A* 113, 1937-1942 (2016). <https://doi.org/10.1073/pnas.1523645113> 27

2. Qiu, X. et al. Induction of photosensitivity by heterologous expression of melanopsin. *Nature* 433, 745-749 (2005). <https://doi.org/10.1038/nature03345>

Response-Figure 17: Effects of melanopsin overexpression on non-image-forming visual function.

(A) Flat mount retina images showing the Tuj1 (green) and melanopsin (red) staining in Ctrl. mice and AAV-melanopsin overexpression (Mela. OE) mice. Scale bar: 50 μ m.

(B) Quantification of the percentage of melanopsin+ RGCs among Ctrl. and Mela. OE groups. Data presents mean \pm SEM. n=3 mice. (***) $p \leq 0.001$, student's t-test)

(C) Representative images of the melanopsin (red) and Tbr1 (green) staining in sectioned retinas showing T-RGCs also expressed melanopsin in the Mela. OE group. Scale bar: 50 μ m.

(D-G) Quantification of the percentage of pupil constriction dynamics within 1 minute of light stimulation (D, F) and the pupil constriction following 1 minute of light stimulation (E, G) in Ctrl. and Mela. OE groups at light intensities of 0.5 mW/cm² (D, E) and 0.01 mW/cm² (F, G). Data presents mean \pm SEM. n=7-8 mice. There were no significant differences between the Ctrl. and Mela. OE groups at any time point in (F). (* $p \leq 0.05$, ** $p \leq 0.01$, ANOVA followed by Bonferroni test in (D) and (F); * $p \leq 0.05$, ** $p \leq 0.01$, student's t-test in (E) and (G))

13. Retrograde Labeling:

Please ensure CTB labeling is confined to the OPN without leaking into neighboring areas like the LGN. Authors should include brain sections showing CTB injections to confirm this.

Response: Thank you for your suggestion. We confirmed that the CTB injection into the OPN did not leak into neighbouring areas (Response-Figure 18). Brain sections demonstrating the CTB injection in the OPN are included in the revised manuscript (see Supplementary Fig. 2A-2B).

Response-Figure 18: CTB injection was confined to the OPN.

(A) Representative images showing the CTB-555 (red) injection sites in OPN regions from Sham and Injury groups, the brain was outlined by DAPI staining (blue). Dashed rectangles outline the OPN regions, which are also enlarged in (B). Arrowheads indicate the lesion site. Scale bar: 200 μ m.

(B) Showing the enlarged rectangle areas in (A). Scale bar: 100 μ m.

14. Stroke Model and CST Sprouting:

While the stroke model and CST sprouting results are interesting, they don't fit with the main focus of the study. Consider removing these results or expanding their characterization to better integrate them with the main work.

Response: Thank you for your suggestion. We decided to remove this part from the revised manuscript.

Reviewer #4 (Remarks to the Author):

Axonal regeneration following axotomy is very poor for CNS neurons in general and retinal ganglion cells (RGCs) in particular. However, several groups, most notably that of Zhitang He, have discovered ways to promote regeneration. First, they enhanced mTOR and JAK/STAT signaling by conditional deletion of their endogenous inhibitors, PTEN and SOCS3, from RGCs. Second, they overexpressed CNTF to further boost signaling. Third, they transected the nerve near the target to avoid the need for long-distance regeneration, thus allowing the RGCs axons to grow enough to form synapses with target cells. Fourth, they used a potassium channel blocker to enhance axonal conduction in the face of deficient remyelination. With these methods, they showed synaptogenesis in the superior colliculus and partial recovery of the optomotor reflex (e.g., refs 6, 10, 23, 53, 54).

Zhang et al. have adopted these strategies, but vary the model by crushing the optic tract in a more convenient place, examining synapse formation in a pretectal nucleus (rather than colliculus), and assaying restoration of the pupillary light (rather than optomotor) reflex. They then use this model to (1) document reestablishment of synaptic connectivity by ultrastructural and physiological criteria, with numerous excellent controls (Figures 1-3D); (2) demonstrate behavioral recovery by showing that the pupillary light reflex is partially restored (Figure 3E-H); (3) provide evidence for differential regeneration by distinct RGC types (Figure 4A,B); and (4) describe transcriptional changes that accompany regeneration. They also show that recovery can be enhanced in three different ways: by Lipin 1 knockdown, melanopsin overexpression and activation of presynaptic calcium channels. These additional accomplishments distinguish this paper from those of He et al., so they not only provide a valuable replication but also provide new insights.

There are, however, several problems that need to be addressed:

1) Much is made of the the advantage of the current “between the LGN and the SC” model over Bei’s “proximal to the SC” model and the yet to the non-specialist they sound very similar. Some explanation, and perhaps a better diagram is needed to indicate the difference.

Along with the advantages of this model there are a few disadvantages. One is that it requires bilateral rather than unilateral axotomy. Recovery also seems slower – it is evident by 3 months after injury in He’s work but only by 6 months here. These are clear from the paper, but should be noted more clearly in the discussion, especially since introduction of the new model is a main point of the paper.

Response: Thank you for your suggestions. We created a new version of the diagram (see Video S1 and Response-Figure 5) to clarify our model. Additionally, we included a discussion on the limitations of our model.

[“the model does have certain limitations, such as the requirement for bilateral rather than unilateral axotomy due to the bilateral projection of RGC axons to the OPN, as well as a

relatively slow recovery process”]

2) “intracranial OTI did not cause noticeable RGC death, in stark contrast with the substantial RGC loss caused by the intraorbital optic nerve crush (ONC).” This dependence of survival with distance has been noted previously, which should be acknowledged.

There is also the complication that not all RGC axons are transected in the model – connections of the LGN and perhaps other retinorecipient areas are maintained. The increased survival is surely in part due to this sparing. In addition, it is possible that intact RGCs can provide support to injured RGCs. In short, these caveats need to be emphasized so the reader will be able to judge how much confidence to place in the conclusion.

Response: Thank you for your suggestions. In the revised manuscript, we have included citations from earlier studies that highlight the dependence of survival on the distance and location of the lesion site. [“the distance from the lesion site and the presence of intact branches before the lesion site appear to be more significant factors, as previously reported^{1,2,3,4*}”].

The striking difference in RGC survival is only partially attributable to the number of RGCs that are injured, as the OTI model affects all axons beyond the LGN, which come from over 90% of all the RGCs, according to the CTB-555 retrograde labeling results (see Supplementary Fig. 1D-1E and Supplementary Fig. 2). However, as the reviewer pointed out, there are still many intact collaterals before the injury site, which may also contribute to RGC survival^{2,4}. We have highlighted these possibilities in the revised manuscript.

1. Berkelaar, M., Clarke, D., Wang, Y., Bray, G. & Aguayo, A. Axotomy results in delayed death and apoptosis of retinal ganglion cells in adult rats. *J. Neurosci.* 14, 4368-4374 (1994).
2. Zheng, B., Lorenzana, A. O. & Ma, L. Understanding the axonal response to injury by in vivo imaging in the mouse spinal cord: A tale of two branches. *Exp Neurol* 318, 277-285 (2019).
3. Lorenzana, A. O., Lee, J. K., Mui, M., Chang, A. & Zheng, B. A surviving intact branch stabilizes remaining axon architecture after injury as revealed by in vivo imaging in the mouse spinal cord. *Neuron* 86, 947–954 (2015).
4. Gao J, Provencio I and Liu X. Intrinsically photosensitive retinal ganglion cells in glaucoma. *Front. Cell. Neurosci.* 16:992747 (2022).

3) The preservation of RGCs also makes the RNAseq data difficult to interpret, since the results are obtained from a combination of axotomized and intact cells. The fraction of “spared” RGCs should be estimated to help the reader understand both the RNAseq and the survival data.

Response: Thank you for your suggestions. We performed retrograde labeling by injecting CTB555 into the SC and OPN of intact mice to estimate the percentage of RGCs projecting axons beyond LGN. The result showed that over 90% of RGCs were labeled with CTB555 (see Supplementary Fig. 1D-1E and Supplementary Fig. 2), which indicates that our OTI model injured over 90% of the RGCs. This indicates that the RGCs we collected were primarily from axotomized cells, with less than 10% originating from uninjured RGCs.

4) Page 13: “Only RGCs from the other eye with axons establishing functional synapses in OPN can be labeled with GFP.” It would take a lot more data to make this statement convincing, since proximity or nonfunctional synapses could also support transfer. This is less convincing than other lines of evidence provided for synaptogenesis – ultrastructure and electrophysiology – and should be toned down or deleted.

Response: Thank you for your suggestion. We agree that nonfunctional synapses may exist between regenerated axons and OPN neurons, potentially leading to PRV transfer. So we revised this sentence to soften the statement by removing the word "functional." ["RGCs from the other eye, whose axons establish synapses with OPN neurons that project to the EW, can be labeled with GFP"]

We would like to keep the results of this part, as it provides an estimate of the composition of RGC subtypes that may contribute to the formation of new synapses and functional recovery.

5) On specific regeneration of ipRGCs. A caveat here is that M1, M2 and alpha-RGCs are the most successful regenerators (Duan, Neuron, 2015; Tran, Neuron, 2019). Although this preference is decreased in the PTEN/SOCS3/CNTF case it is not abolished (Jacobi, Neuron 2022). So a caveat here is that the selectivity may be partly authentic but also partly a reflection of different regenerative ability.

Response: Thank you for your comment. To clarify from the outset, our PRV-GFP retrograde tracing data does not identify all types of RGCs that regenerate their axons; it only identifies the cell types that form synapses with the OPN neurons projecting to the Edinger-Westphal nucleus (EW). This is because PRV-GFP was delivered to the anterior chamber of the eye to infect the iris, which is downstream of OPN in the PLR circuit.

In the PTEN/SOCS3/CNTF case (Jacobi, Neuron 2022), M1-M2 ipRGCs and alpha-RGCs constituted less than 36% of all the regenerated RGCs. Since we utilized a similar genetic strategy, we anticipate that many other RGC subtypes, in addition to M1-M2 ipRGCs and alpha-RGCs, regenerated their axons to cross the lesion site. However, our study did not focus on which RGC subtypes regenerated better, as other groups have already addressed this in the optic nerve crush model. Instead, using our OTI model, we are more interested in the RGC subtypes that re-establish synapses with target neurons related to functional recovery. Therefore, we conducted trans-synaptic tracing experiments using PRV-GFP, and the labeling of ipRGCs with PRV-GFP does not indicate the differing regenerative abilities of RGC subtypes. However, we cannot rule out the possibility that in our case, ipRGCs and alpha-RGCs were still the major RGC types that regenerated into OPN.

6) Effects of Lipin knockdown seem noteworthy. However, this reviewer knows nothing about lipin, and no explanation is provided. Background is required – what does it do, what effects on regeneration have been demonstrated previously, why was it chosen and so on. In addition,

no data are provided on the efficacy and specificity of lipin knockdown, making it impossible to judge the result.

Response: Thank you for your insightful comment. We apologize for not including key information about Lipin1 in the previous manuscript. Below, please find the background details about Lipin1, which have also been incorporated into the Results section. The efficacy of Lipin1 knockdown was demonstrated in our previous work, which was also added to the manuscript Methods “AAVs packaging” section.

[Lipin1 is a phosphatidic acid phosphatase (PAP) enzyme that is essential for the conversion of phosphatidic acid to diacylglycerol in the glycerol phosphate pathway. Previous research has demonstrated that the depletion of lipin1 redirects lipid metabolism from triacylglyceride to phospholipid synthesis and enhances the mTOR and STAT3 signaling pathways to facilitate axon regeneration. Additionally, axon regeneration was synergistically improved by combining Lipin1 knockdown with PTEN deletion or CNTF overexpression following optic nerve injury.]

6) Regarding the lipin result – it is not clear that regeneration is better in the PSL condition (Fig. 5) than when lipin is unaffected (Fig.1) These conditions need to be compared in the same experiment. Similarly, for restoration of function, PSCL needs to be compared with PSC in the same experiment since there may be a rapid recovery in the former between 3 and 6 months. In other words, is it possible that PSC recovers in 3.5 months?

Response: Thank you for your suggestions. We compared PLR recovery between the PSCL and PSC groups in the same experiment and found that only the PSCL group showed PLR recovery at 3 months post-injury, while the PSC group did not exhibit any evident PLR recovery until 6 months post-injury. Specifically, the PSC group did not show recovery at 4 months. Although some mice with PSC treatment recovered their PLR within 5 months, the majority did not (Response-Figure 19).

While we did not directly compare regeneration between the two groups in the same experiment, regenerated axons were observed growing into the OPN at 3 months only in the PSCL mice, not in the PSC mice. These findings suggest that axon regeneration was better in the PSCL condition.

Response-Figure 19: Lipin1 knockdown combined with Pten/Socs3 knockout and CNTF expression (PSCL) significantly accelerated PLR functional recovery after pre-OPN OTI. Unlike the PSCL Regen. group, which exhibited PLR recovery by 3 months post-OTI, no PLR responses were observed in the PSC Regen. group until 5 months post-OTI, with only 1 out of 5 mice responding. PLR recovery was consistently observed in all PSC Regen. mice by 6 months post-OTI. Data presents mean \pm SEM. n=5 mice. (* $p \leq 0.05$, ** $p \leq 0.01$, ANOVA followed by Bonferroni test)

7) Discussion page 12, lines 17-36. Conclusions about mechanism – for example attraction vs repulsion or roles of genes whose expression has not been validated – are stated with excessive confidence. These are interesting speculations, and need to be presented as such.

Response: We agreed that we should not state with excessive confidence since we have little supporting evidence. We have rephrased our writing as following: “The majority of the identified guidance receptors are from the immunoglobulin superfamily of cell adhesion molecules, which includes Cadm2, Cntn1, L1cam, Ncam1, Ptpns, and Thy1 (Supplementary Fig. 15H). The roles of these molecules in post-injury axonal regeneration and reconnection, as well as functional recovery, require additional experimental investigation.”

REVIEWERS' COMMENTS

Reviewer #1 (Remarks to the Author):

In this revised manuscript, the authors have addressed all of the concerns I raised during previous review. This manuscript will be a great addition to the field of axon regeneration.

Response: Thank you for your recognition and positive feedback on our work.

Reviewer #2 (Remarks to the Author):

The authors have addressed all comments and concerns satisfactorily.

One suggestion remains - to please add sufficient details for the surgical procedure as this is a very good model that will be useful to the field.

The schematic added is good, and it appears they crush in the anteroposterior plane. However, the new details mentioned are unclear as the anteroposterior coordinate is -3.6mm. Presumably the crush injury extends from -0.6 to -3.6mm, i.e. the two arms of the forceps?

Response: Thank you for your positive feedback and suggestions regarding our work. We have included additional details about the surgical procedure in the Methods section as recommended:

“During the surgery, the forceps were fixed to the stereotaxic holder with a 3 mm distance between the two tips. The left tip of the forceps was gently placed on the top of the bregma, which was set as the zero point. The left tip was then moved to the coordinates (anteroposterior -3.6 mm, mediolateral ± 1.7 mm), while the right tip was positioned at anteroposterior -0.6 mm and mediolateral ± 1.7 mm. Bilateral optic tract crush was then performed at a depth of 3.5 mm for 5 seconds.”

Reviewer #3 (Remarks to the Author):

This is a very well done revision. Congratulations!

Response: Thank you for your recognition and positive feedback on our work.

Reviewer #4 (Remarks to the Author):

The authors have provided an outstandingly comprehensive set of revisions in response to the comments of the reviewers. I congratulate them. Their answers and revisions address my concerns (as reviewer 4) and as far as I can tell, those of reviewers 1-3.

I have three very small suggestions.

1. (related to my point 3) In discussing the role of surviving axons on the RNAseq results, they note that only 10% survive. This is a low but not insignificant number. It should be noted in the text.

Response: Thank you for your positive feedback and suggestions regarding our work. As you pointed out, approximately 10% of RGCs were not injured by our OTI model. We have added a discussion on this in the revised manuscript to clarify its potential impact on our findings. "This difference is partially explained by the fact that the OTI model leaves approximately 10% of RGCs uninjured. More importantly, the varying distances of axonal injury from the cell body and the presence of intact branches likely play a more significant role in shaping the distinct signaling responses."

Additionally, we also noted this in the Results section: "The striking difference in RGC survival is only partially attributable to the number of RGCs that are injured, as the OTI model affects over 90% of RGCs, according to the CTB-555 retrograde labeling results (Supplementary Fig. 2D-2E). Instead, the distance from the lesion site and the presence of intact branches before the lesion site appear to be more significant factors, as previously reported".

2. (related to my point 5) In discussing the selectivity of synapse formation, they note that only 36% of the regenerating axons are ipRGCs or alpha RGS whereas, they see closer to 80% in these categories. There is a clear difference as they note. However, the 36% is already an order of magnitude enrichment over the population as a whole, so this number and the comparison should be noted in the text.

Response: Thank you for your suggestions. Our focus is not on which RGC subtypes regenerate better, but rather on those that re-establish synapses with target neurons related to functional recovery. The percentage of identified RGCs highlights a clear difference between the two scenarios: 88% of the RGC subtypes that re-establish synapses with target neurons in our experiment are ipRGCs, while less than 36% of the RGC subtypes that regenerated their axons in the study by Jacobi et al. (Neuron 2022) are ipRGCs and alpha-RGCs. Since the two studies investigated RGC composition under different scenarios and used different injury models (we used optic tract injury while they used optic nerve injury), a direct comparison of these percentages may lead to misunderstandings for readers. Instead, we believe that emphasizing the different research questions will provide clearer context, so we have added a discussion about this.

"We did not specifically investigate which RGC subtypes regenerated better during the axon regeneration process, though we anticipate that many other RGC subtypes, in addition to M1-M2 ipRGCs and alpha-RGCs, could regenerate their axons across the

lesion site according to a study using similar strategy to promote axon regeneration in ONC model”.

3. The figures for the reviewer are very helpful. I’m having trouble figuring which ones cover material that is also described or illustrated in the article. If there are a few additional panels that could be added to the main or supplementary figures, I’d be very supportive.

Response: Thank you for your suggestions. We have included additional figures that were not present in the previous revised manuscript in the latest version. You can find these in Supplementary Fig 11E, and Supplementary Fig 14A.